# The leptomeninges as a critical organ for normal CNS development and function: First patient and public involved systematic review of arachnoiditis (chronic meningitis)

Carol S. Palackdkharry[1][*], Stephanie Wottrich[2][¤], Erin Dienes[3], Mohamad Bydon[4], Michael P. Steinmetz[5], Vincent C. Traynelis[6]

1 Arcsology®, Saint Charles, Illinois, United States of America, 2 Case Western Reserve School of Medicine, Cleveland, Ohio, United States of America, 3 Arcsology®, Mead, Colorado, United States of America, 4 Department of Neurologic Surgery, Orthopedic Surgery, and Health Services Research, Mayo Clinic School of Medicine, Rochester, Minnesota, United States of America, 5 Department of Neurological Surgery, Cleveland Clinic Lerner College of Medicine Neurologic Institute, Cleveland, Ohio, United States of America, 6 Department of Neurosurgery, Rush University School of Medicine, Chicago, Illinois, United States of America

☯ These authors contributed equally to this work.
¤ Current address: Department of Neurology, University of Texas Dell Seton Medical Center, Austin, Texas, United States of America
* dr.palackdharry@arcsology.org

## Abstract

### Background & importance

This patient and public-involved systematic review originally focused on arachnoiditis, a supposedly rare "iatrogenic chronic meningitis" causing permanent neurologic damage and intractable pain. We sought to prove disease existence, causation, symptoms, and inform future directions. After 63 terms for the same pathology were found, the study was renamed *Diseases of the Leptomeninges (DLMs)*. We present results that nullify traditional clinical thinking about DLMs, answer study questions, and create a unified path forward.

### Methods

The prospective PRISMA protocol is published at Arcsology.org. We used four platforms, 10 sources, extraction software, and critical review with ≥2 researchers at each phase. All human sources to 12/6/2020 were eligible for qualitative synthesis utilizing R. Weekly updates since cutoff strengthen conclusions.

### Results

Included were 887/14286 sources containing 12721 DLMs patients. Pathology involves the subarachnoid space (SAS) and pia. DLMs occurred in all countries as a contributor to the top 10 causes of disability-adjusted life years lost, with communicable diseases (CDs) predominating. In the USA, the ratio of CDs to iatrogenic causes is 2.4:1, contradicting arachnoiditis literature. Spinal fusion surgery comprised 54.7% of the iatrogenic category, with

**Data Availability Statement:** Table 5 contains all included baseline data and associated references. The complete table, with all data and references, is

available from our public Arcsology site (https://arcsology.org/the-complete-sr-dataset).

**Funding:** No researchers were compensated. Software and ongoing reference repository fees were provided by contributions to Arcsology,® a 501c3 organization. Open access publication fees provided by the A. Watson Armour and Sarah Armour Presidential Endowed Chair of Vincent C. Traynelis MD.The funders had no role in study design, data collection and analysis, decision to publish, or preparation of the manuscript.

**Competing interests:** Dr. Palackdharry: none Stephanie Wottrich: none. Dr. Dienes: none. Dr. Bydon: none Dr. Steinmetz: Elsevier: royalties; Globus: consultant; Medtronic: honorarium; Zimmer Biomet: royalties. Dr. Traynelis: Medtronic: consultant, royalties; NuVasive: consultant This does not alter our adherence to PLOS ONE policies on sharing data and materials.

rhBMP-2 resulting in 2.4x more DLMs than no use (p<0.0001). Spinal injections and neuraxial anesthesia procedures cause 1.1%, and 0.2% permanent DLMs, respectively. Syringomyelia, hydrocephalus, and arachnoid cysts are complications caused by blocked CSF flow. CNS neuron death occurs due to insufficient arterial supply from compromised vasculature and nerves traversing the SAS. Contrast MRI is currently the diagnostic test of choice. Lack of radiologist recognition is problematic.

## Discussion & conclusion

DLMs are common. The LM clinically functions as an organ with critical CNS-sustaining roles involving the SAS-pia structure, enclosed cells, lymphatics, and biologic pathways. Cases involve all specialties. Causes are numerous, symptoms predictable, and outcomes dependent on time to treatment and extent of residual SAS damage. An international disease classification and possible treatment trials are proposed.

## Introduction

### Objectives

This is a unique global patient and public involved (PPI) study. The topic was contentious when we started in 2018. Lawsuits had been filed and settled. Certain procedures or devices even made a Consumer Reports issue [1]. The PPI—multi-institution neurosurgery team objective was to perform an exhaustive prospective PRISMA systematic review (PSR) to address a presumed iatrogenic disease called arachnoiditis causing permanent disability and intractable neuropathic pain (NPP) after procedures for common events such as childbirth or back pain. Chronic meningitis (CM), the original but non-specific name, has the same International Classification of Diseases (ICD) 10 code, thus was included. After 63 disease names for the same inflammatory or fibrotic subarachnoid space (SAS) and pia pathologies was reached, a major protocol modification changed the study name and objective to *Diseases of the Leptomeninges* (DLMs), doubling search returns.

### Rationale

The PPI group was composed of >20 medical, scientific, and biostatistical doctorates, as well as nurses and other clinicians trained in systematic review methodology, representing >15K Facebook (FB) members with arachnoiditis as either patients themselves or family. No quality published reviews on arachnoiditis existed and only three reputable internet sources were available from USA search engines in 2018: the National Organization for Rare Diseases [2]; the National Institute of Neurologic Disorders and Stroke [3]; and the Genetic and Rare Diseases Information Center [4]. All described a rare, iatrogenic disease of the arachnoid membrane with no known treatment, resulting in progressive intractable NPP and neurologic loss.

Five pre-planned steps made the seminal conclusions from this PSR possible: (1) using internet search engines to find additional terms for arachnoiditis and chronic meningitis prior to creating structured searches; (2) hand searching and capturing alternative terminology; (3) understanding biases and limitations of published literature and incorporating additional resources; (4) concurrently capturing animal and molecular studies in a separate database to inform future directions; and (5) the study team *experiencing*, *causing*, or *witnessing* the same

unusual set of symptoms reported by 15K FB patients with arachnoiditis documented in source after source for 200 years. Fig 1 shows the PRISMA diagram.

DLMs go by yet another name in the spinal intervention space, explaining the difficulty finding information about this "iatrogenic disease." In 1978 neurosurgeon Charles Burton MD, founder of the North American Spine Society, recommended the term "radiculitis" to differentiate acute arachnoiditis from "adhesive arachnoiditis" as the cause of new neurologic symptoms immediately postoperative or post-myelogram [5]. Burton demonstrated the surgical appearance of each entity with operative photographs (Fig 2). This term still predominates in the spine surgery and procedure spaces.

Quantifying iatrogenic harm is known to be veiled in most publications. Author—commercial ties may impact willingness to report negative outcomes from sponsored researchers [6]. Adverse events (AE) from drug and medical devices are not accurately reflected in the medical literature due to lawsuits settled with confidentiality orders [7,8]. AE reported to both public and hidden government databases are not indexed. Thus, we also included case reports, government sources, legal literature, websites, investigative reporting, and other prospectively defined grey literature.

Although arachnoiditis is also called chronic meningitis, a 2021 expert review of chronic meningitis does not mention arachnoiditis, stating "general statements about prognosis are difficult to make" [9]. However World Health Organization (WHO) data demonstrate frequent permanent disabilities from arachnoiditis, many resulting from communicable diseases (CDs), accounting for $\geq 2$ of the top 10 causes of disability-adjusted life years lost (DALY) in most countries [10]. Furthermore, specific neurotrophic infection-related causes of permanent neurologic damage generally are referred to by a causative organism name. Two of many examples are TBM (tuberculosis meningitis) and neuroborreliosis (NB) for Lyme disease neurologic manifestations, with neurologic damage from both classified as arachnoiditis.

Little has changed since study inception. Weekly data updates since the cutoff date further strengthen results. The newest WHO data show in South Africa, sequelae of DLMs from HIV, TBM, interpersonal violence, and road accidents are all in the top 10 causes of DALY. In the USA, injury from violence and "legal intervention" is number six and sequelae from meningitis is number 10.

We synthesized $> 60$ different LM disease names into a spectrum of organ damage, thereby presenting a cohesive picture of the clinical impact of DLMs throughout life. These results nullify previous clinical teaching about the LM and for the first time, clinically verify laboratory neuroscience advances.

## Methods

### Patient-public involvement

PPI permeated this study. The 17-member e-Delphi group included nine patients and caregivers: eight doctors (six physicians, one molecular geneticist, 1 biostatistician), and a computer engineer. The nine PPI e-Delphi members were engaged in every aspect of the study and two are authors. Five FB group leaders surveyed their collective 15K members and prospectively collated specific patient questions and symptoms to research (Table 1).

### Eligibility criteria and information sources

Table 2 summarizes eligibility requirements, search modifications, databases used, included grey literature, and lists all disease terms with proven leptomeningeal (LM) pathology. Full text sources, including "Unpaywall" are shown [11]. Qualitative synthesis utilized data from database inception through the 12/6/2020 all-database search. Weekly updates of PubMed

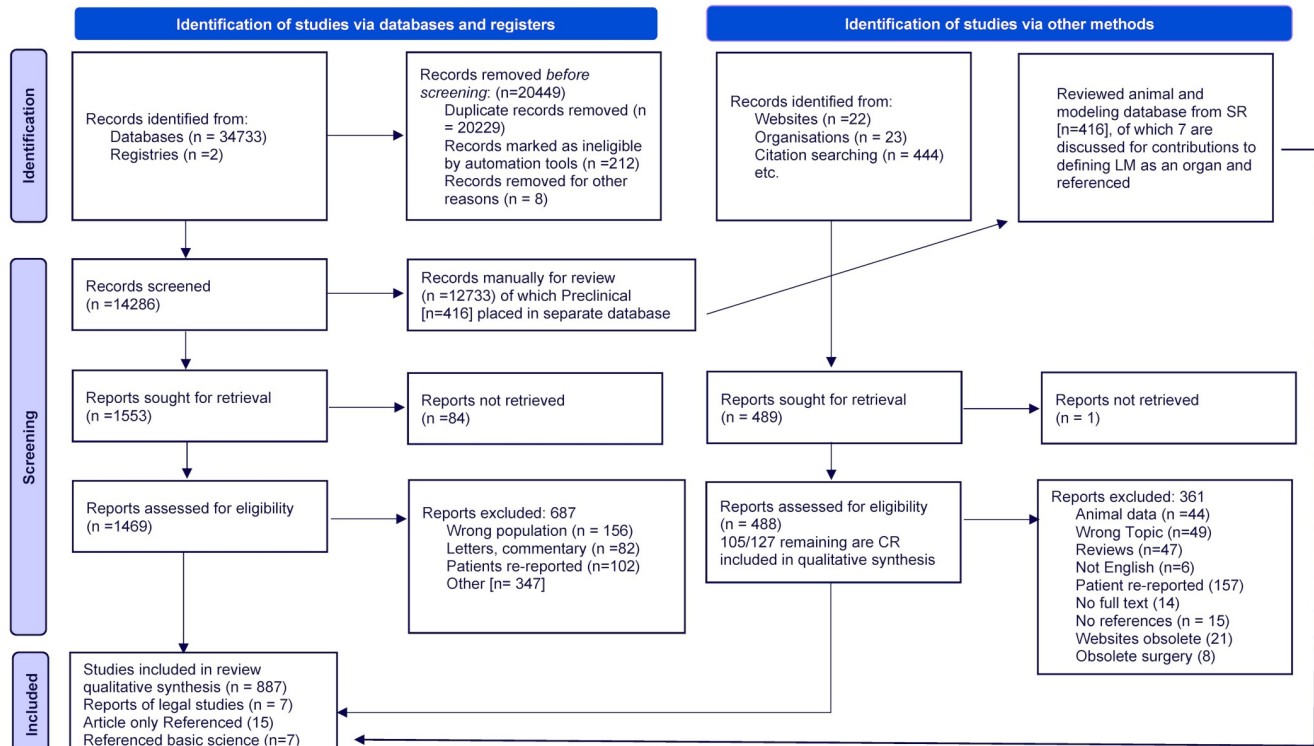

**Fig 1. PRISMA-2020 flow diagram.** Diagram includes searches of databases, registers and other sources, and pre-clinical studies for leptomeningeal pathology review.

searches are current through 7/6/2022, with new reports further confirming synthesized data. Inability to obtain free full text resulted in exclusion. No full text was retrievable in 73/1913 (3.8%) articles; 70 were case reports published prior to 1980 and the remaining three dated before 1948.

## Search strategies

The entire PSR protocol, including structured searches, returns, and modifications, is published at Arcsology.org [12]. Prospero did not register this complicated methodology in 2018. Alternative terminologies for DLMs were captured in a software text field during screening and hand searching.

After finding 34 disease names not picked up by search#1, despite pathology demonstrating involvement of the subarachnoid space (SAS) and pia, search#2 was added and protocol modification in April 2019 changed the study name to DLMs. Members of the PPI group performed Google and Google Scholar searches.

## Source selection

The PSR Flow 2020 diagram is shown in Fig 1. Sources were evaluated by ≥2 independent clinicians at every phase of the PSR using the inclusion, exclusion, and quality criteria shown in Tables 2 and 3. A standardized process for disagreement resolution was used. Each patient was knowingly included only once, thus multiple publications on the same patient population were not eligible.

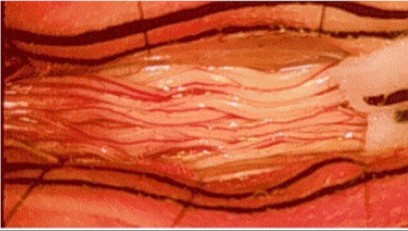
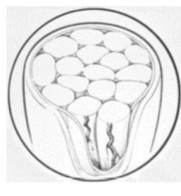

Photograph at surgery of swollen and hyperemic nerve fibers which have herniated out of the dural Incision to fill the operative field, Indicating the previous Increased tension within the dura. Radiculitis phase of lumbosacral arachnoiditis

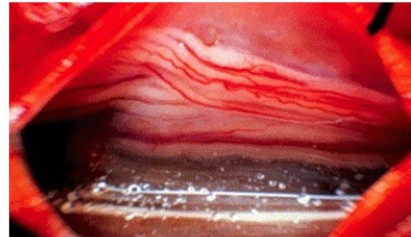
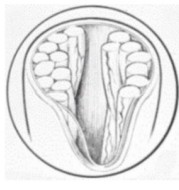

The opened dura shows nerve fibers of the cauda equina held by collagenous scar tissue to each other and to the dura . Small globules of Pantopaque are floating In the cerebrospinal fluid at the bottom. Arachnoiditis, phase of lumbosacral arachnoiditis.

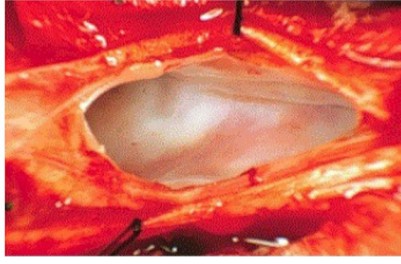
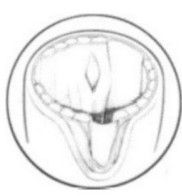

Opening of the dura exposes an apparently empty cavity. The atrophied nerve roots are enmeshed in solid collagenous scar tissue and are plastered to the dura and to each other. Thinned nerve roots are seen running along the edges of the dural opening. Adhesive arachnoiditis, phase of lumbosacral arachnoiditis.

**Fig 2. Neurosurgeon Charles Burton's operative pictures of different phases of pathology after Pantopaque myelogram and/or spine surgery.** (5) Each section contains an operative picture of his proposed 3 phases, followed by his illustration and description. The first section shows nerve inflammation of the cauda equina, which he termed "radiculitis" to differentiate this stage from "adhesive arachnoiditis" shown in the second panel. Burton proposed adhesive arachnoiditis as a middle phase with clearly visible nerve root clumping and early nerve atrophy. The final phase, often termed "empty thecal sac" appearance on MRI is characterized by nerve roots permanently adhered to the dura and covered with fibrotic tissue.

Given the presumption of iatrogenic causes of arachnoiditis in 2018, the dramatic increases in rhBMP-2 (recombinant human bone morphogenetic protein 2), and widely questioned data integrity and AE in initial studies, we closely examined rhBMP-2 as a possible contributing source. One of the two manufacturer-commissioned YODA (Yale Open Data Access) systematic reviews attempting to address safety was included [13,14]. All rhBMP-2 studies not included were eligible, as was a Food and Drug Administration (FDA)) AE report released to the Star Tribune in 2016 through the Freedom of Information Act which adds AE to spinal fusion surgery results, challenging rhBMP-2 safety [15].

The database reportedly contains 1039 postoperative complications in 3647 patients undergoing cervical, thoracic, or lumbosacral spinal fusions with rhBMP-2 from 26 study sites but is not pristine due to "batching" of later AE. The AE were extracted in duplicate per protocol. All authors reviewed the data, with substantial neurosurgeon debate on abbreviated terminology used in portions of the file. There was unanimous e-Delphi agreement to include 632/1039 AE patients with documented new postoperative motor or sensory deficits, most also with new NPP consistent with radiculitis. Unanimous agreement was not reached on an additional 236/1039 AE patients with only new neurologic pain due to vague documentation in latter parts of the file, thus these patients were not included as neurologic AE.

Although this is a deviation from protocol which allowed inclusion if ≥75% e-Delphi approval, due to the impact of this body of data on results, we required a unanimous vote for inclusion. We used protocol-dictated critical review parameters for all other iatrogenic causes of DLMs such as myelogram dye, intrathecal and intraventricular chemotherapy, neuraxial

**Table 1. Global patient responses regarding arachnoiditis questions or associated complications.**

| Patient generated questions | |
|---|---|
| Acute arachnoiditis | Association with smoking, diabetes, drugs, opioids, socioeconomic factors |
| Autoimmune diseases | Autopsy findings |
| Bacterial infection/meningitis, infectious meningitis | Blocked CSF flow, intra-cranial hypertension |
| Bone morphogenetic protein | Carcinomatous meningitis, intrathecal chemotherapy |
| Cauda equina syndrome, neurogenic colon, neurogenic bladder, | Cause, etiology |
| Charcot Marie Tooth Disease | Chemical toxicity |
| Collagen vascular disease SLE, RA, AS, Systemic sclerosis | CRPS Chronic Regional Pain Syndrome |
| CSF leaks | Diagnosis |
| Differential diagnosis | Dural tears, dural patches, fibrin glue, |
| Ehlers Danlos Syndrome | epidemiology |
| Epidural abscess | ESI, epidural steroid injections, selective nerve root blockade |
| Experimental models, animal (mouse, rat, dog) | Fungal infection/meningitis |
| Genetic arachnoiditis, genetic associations, ethical or cultural associations | Glial cell changes / involvement |
| Immunotherapy, anti-inflammatories, anti-inflammatory treatment | In vitro model |
| Low dose naltrexone | Lyme |
| LP, regional anesthesia, epidural anesthesia | MRI appearance, CT myelogram, myelogram |
| Molecular mechanism | Neuropathic pain |
| Neuroinflammation | Oil-based myelogram |
| Neuroregeneration | Pathology, mechanism, glial or microglial changes |
| Outcome, resolutions, progression, natural history, prognosis, course | Prevention |
| Pathophysiology | Resolution, cure |
| Rehabilitation, impact of exercise, disability | Scoliosis |
| Sarcoidosis | Spinal arachnoid cyst: added 1/13/19 |
| Sexual dysfunction, impotence | Stem cell therapy |
| Staging | Surgery |
| Sub-arachnoid hemorrhage SAH | TB |
| Tarlov Cyst added 1/13/19 | Trauma, injury |
| Tick-borne diseases | Ultrastructural changes and sub searches |
| Treatment, immunotherapy, immunosuppression | Viral infection/meningitis |

anesthesia, and spinal injections. Only studies and data passing critical review were included, as discussed below.

## Data collection

Zotero is the reference repository [16]. Human studies were imported into Covidence 1.0 software for full text review and extraction [17]. Due to software limitations, studies were sorted into major and subcategories prior to extraction. Standardized data forms were created for each category through an intensely iterative process. Covidence manually configured and exported this rich body of data.

Apart from Covidence's workarounds, all authors and the Chief Data Integrity Officer had continual database oversight and attest to the integrity of the data.

**Table 2. Eligibility criteria, modifications, information sources, and different terms for same pathology.**

| Protocol Information | Details |
|---|---|
| Inclusion for total search | 1. Not limited to humans. English or English translation available. Any age, sex, gender identity, race, geographic area, and co-existing illnesses were included. |
| Inclusion Criteria for data synthesis | 1. Humans only for the qualitative syntheses. (non-human studies removed by manual review and placed in a separate database). All other factors the same as above<br>2. ≥2 independent reviewers were used for all phases of the review, including quality and bias<br>3. No start date limitation |
| Included grey literature | 1. Case reports in searched databases<br>2. Google: Legal documents, class action lawsuits, government proceedings and databases, manufacturer device adverse events (AE) tracking, investigative reporting, EU proceedings mandating recertification of medical devices to exclude those without clinical data, verifiable website data |
| Exclusion Criteria for data synthesis | 1. Acute meningitis without adequate outpatient follow-up<br>2. Sources with data errors unexplained after contacting authors<br>3. Sources requiring payment for full text despite sources listed below<br>4. Websites claiming unpublished positive results, but data not shared for review<br>5. Published after 1/15/22 for discussion and 12/6/20 for data synthesis<br>6. Critical review: Methods unsound, flawed, clearly biased, but cited as to reason for exclusion |
| Search Modifications | 1. Added structured search for leptomeningeal, arachnoid, subarachnoid, pia fibrosis. Obscure terminologies searched separately added by hand<br>2. Cutoff date modifications: moved from 12/31/2018 to 3/20/19 to 12/6/20 to 5/30/21 (5/30/21: 15 new abstracts, 10 infectious CR (including SARS-CoV-2), 1 autopsy, 4 CS. Links to abstracts added to full online protocol. No data change).<br>3. Addition of ICD 11 terminology and proposal already submitted to WHO Proposal #2C3P<br>4. Change to version 2020 PRISMA which allows for multiple types of input sources 7/1/21 |
| Information sources and databases: | 1. Platform 1: PubMed/NCBI, FDA MAUDE, Global Government registries<br>2. Platform 2 (Mayo Clinic Librarian): Ovid MEDLINE(R) and Epub Ahead of Print, In-Process & Other Non-Indexed Citations, Ovid Embase, Ovid Cochrane Central Register of Controlled Trials, Ovid Cochrane Database of Systematic Reviews, and Scopus.<br>3. Platform 3–4: Google Scholar, Google, Zotero search engine, Endnote plug-in |
| Full Text Sources: | 1. Academic Libraries: Mayo Clinic, Cleveland Clinic, Case Western Reserve, University of Chicago, Rush University<br>2. Apps: Zotero plug-in, Google Scholar plug-in, "Unpaywall" app<br>3. Author contact requesting full text if email was provided |
| Different names for this same LM pathology found in hand searching.*<br>Bolded names (21) contain "arachnoiditis" and non-bolded (42) do not. Listed alphabetically | **adhesive arachnoiditis**, arachnoid adhesions, arachnoid cysts, arachnoid fibrosis, arachnoid scarring, arachnoid ossificans, **arachnoiditis ossificans**, arachnoid webs, **arachnoiditis**, arachnoidopathy, arachnopathy, aseptic meningitis, Bannwarth's Syndrome, basilar fibrosis, basilar meningitis, **cerebral arachnoiditis**, **chemical arachnoiditis**, **chronic circumscribed arachnoiditis**, **chronic circumscribed cystic arachnoiditis,** chronic leptomeningitis, chronic meningitis, **chronic serous arachnoiditis**, **chronic spinal adhesive arachnoiditis**, cystic meningitis, **cystic arachnoiditis**, encephalomeningoradiculopathy, encephalomyeloradiculitis, **epiduro-arachnoiditis**, **familial adhesive arachnoiditis**, **focal adhesive arachnoiditis**, **hereditary arachnoiditis**, leptomeningeal adhesions, leptomeningeal fibrosis, leptomeningeal inflammation, leptomeningeal scarring, leptomeningitis, meningeal fibrosis, meningeal scarring, meningeal adhesions, meningitis, meningitis serosa, meningitis serosa circumscripta spinalis, meningoradiculitis, meningoradiculopathy, myeloradiculitis, myeloradiculopathy, **neoplastic arachnoiditis**, **optochiasmatic arachnoiditis**, post-op radiculitis (current spine literature), **postmyelographic arachnoiditis**, Pseudotumor cerebrii, radiculitis, radiculomyelitis, **rhinosinusogenic cerebral arachnoiditis**, serosa circumscripta spinalis, **spinal arachnoiditis**, subarachnoid cysts, subarachnoid fibrosis, syringomyelia, Tubercular meningitis, Tuberculosis meningitis, ventriculomeningitis |

*A title search for arachnoiditis would have returned 21/63 of the terms used in the literature (appearing in bold). All the terms describe some form of leptomeningeal disease, with or without the involvement of the underlying CNS. Most papers had surgical descriptions and pictures, pathology, or autopsy verification of leptomeningeal disease.

**Table 3. Critical appraisal of sources.** These questions were used in our quality and bias tool.

| Bias parameter | General Bias Comments | Bias resulting in exclusion noted |
|---|---|---|
| Case studies | Intrinsic known biases | None |
| Bias due to financial interest? | 1. spinal device and biologics: most studies appeared to have industry sponsorship<br>2. Many spinal device companies "provide money to the institution" of authors<br>3. several studies of new devices and techniques that theoretically could lead to financial gain<br>4. Oncology Phase 1 and 2: most studies had industry funding and assistance with data | 1. One hrBMP trial excluded due to high bias: industry sponsorship + additional money paid to authors + manufacturer analyzed the data and helped write paper. |
| Adequate follow-up (18 months)? | 1. Most spinal intervention trials ranged from 24 hours to a max of 6–12 month follow up<br>2. Same is true for most acute meningitis studies, which is why this was not excluded in searches | 1. No studies were excluded on this factor alone, understanding in all the spinal intervention studies, follow up for LMD was inadequate |
| Was sponsoring company involved in data analysis and writing of study? | 1. Unclear in many studies prior to mandatory COI disclosures.<br>2. In oncology, rhBMP-2, and new neuroendoscopy device trials, manufacturers involvement in data analysis was not uncommon, but involvement in paper writing was rarely disclosed | 1. One study excluded, as noted above |
| Bias in testing or outcome due to lack of resources (or year of study)? | 1. Due to the 100-year span of sources, there was significant differences in available technology and resources in all countries<br>2. Similar situation for myelogram and infectious studies prior to current technology and treatment<br>3. Current issues in resource limitations in N-OECD are notes as MRI not universally available | 1. None, but noted as outcomes may be tied to early detection and treatment, especially in infectious cases |
| Is COI declared? | 1. Not required or present prior to past 10–15 years, so many papers had no COI declared.<br>2. Multiple more recent studies from other countries translated into English did not contain COI. | 1. None based on this factor alone. In current studies with multiple authors, the COI declarations can be longer than the actual article. Clearly industry is funding many medical advances and education opportunities. |
| Evidence of selective outcome bias? | 1. Several device and invasive studies reported positive outcomes, but did not completely address negative outcomes or longer-term outcomes<br>2. Spinal anesthesia and non-surgery spinal interventions (radiology or pain management) generally did not report on AE occurring after discharge from the hospital<br>3. Oncology study outcomes were exceedingly difficult to determine as the definition of LMD was often absent or not consistent with neurologic-specialty definitions | 1. One study noted above did not have stated follow up for >50% of patients included.<br>2. A second study was excluded due to a methodology that resulted in high selective outcome bias resulting in altered mortality rates<br>3. A third study on the role of physical therapy was excluded do to complete lack of objective evidence or physician exams |
| Do inclusion/exclusion represent real-world patients? | 1. Spinal surgery without devices or biologics generally reported 6–12 months of follow up<br>2. Invasive spinal interventions resulting in adverse events appear to be vastly under-reported as supported by multiple sources and the discussion of legal settlements with non-disclosure clauses.<br>3. Oncology studies of giving intra-CNS therapy appears to need further research on the impact of intraventricular vs. intrathecal therapy as these therapies do not have equivalent CSF flow distribution | A 4th study was removed after detailed review of the ICD 9 coding methodology used. Several important codes for additional complications were missing |
| Reviewer bias? | 1. One unique aspect of this ScR is that the primary first reviewers are clinicians that have a diagnosis of arachnoiditis with each having a bias towards proving what caused our diseases<br>2. At the same time, we had to achieve consensus with neurosurgeons and neurologists from 5 institutions with a possible bias towards disproving those causes.<br>3. Reviewers were asked to declare their personal biases | Several studies were excluded early in the PRISMA process due to single reviewer bias with our process of 3 blinded reviewers. None were excluded for this reason in the final stages, with multiple checks and balances in place. |

## Data items

Table 4 lists collected data elements and timeframes. The attempt to assess individual social determinants of health (SDOH) proved futile as they were rarely presented. We collected and used the Organization for Economic Co-operation and Development (OECD) status as a

**Table 4. Data elements captured.** Standard data elements were captured for all sources if present. Specialized baseline characteristics were charted on in the listed population, in addition to standardized elements.

| Type of data element | Data captured |
|---|---|
| Standardized study identification | Sponsorship source; country; setting; first author name; institution; contact email; journal, year of study |
| Standardized: study issues | Corrections, withdrawal, significant discussion in editorial or letters |
| Standardized: other details | OECD vs. N-OECD; category (i.e., treatment, causes, etc.); source (if not journal); additional names being used to describe arcs or disease of leptomeninges |
| Standardized free text population data | Inclusion and exclusion criteria; use of alternative therapies documented; how definitive diagnosis was made |
| Standardized: demographics | Mean/range age; % male; # patients; SDOH if given (tobacco, BMI, diabetes, at risk for low Vit D level); duration of follow-up |
| Standard: treatment (non-chemotherapy) | Intervention, positive clinical response, negative clinical response, length of response, duration of follow up after intervention, extent of disability over time, non-clinical documentation of response, steroids also used |
| Standardized outcome: symptom improvement over time in those who had symptoms | Symptoms of hydrocephalus; vision; cranial nerves (total, other than vision); cognition; neuropathic pain; hypesthesia or sensory loss; abnormal reflexes; motor dysfunction; muscles spasms or fasciculation; bowel, bladder, or sexual dysfunction; cerebellar or posterior column; severe adjustment disorder or suicidal ideation. Measured at baseline, 1m, 3m, 6m, 1y, 2-4y, 5-10y |
| Standardized: general outcome | Progression despite intervention; death; lost to follow-up; ability to return to work if under 65, leisure activities, or school. |
| Standardized outcome: care delay | Time to: development of clinical symptoms after triggering event; obtain MRI or diagnostic test after development of symptoms; obtain adequate palliation of symptoms and pain control after diagnosis |
| Additional: infections | Agent; time from exposure to symptoms; CSF standard; imaging; how diagnosis made (antibodies, culture, peripheral blood antibodies, next gen sequencing, etc.); HIV status, chemotherapy used and response; surgery needed and response; areas involved (brain, spine, both), short term response, long term sequela, any IT therapy used, steroids used |
| Additional: cancer, with Toxicity grading system explanation for cancer trials (including intra-CNS therapy) | Pathology; prophylaxis vs CNS disease; CSF; any radiation therapy to brain or spine; chemotherapy given; route of therapy (intraventricular vs intrathecal); number of therapies; development of arcs using correct arcs definitions, outcome; use of steroids, use of immunostimulation and arcs neurotoxicity |
| | CTCAE Grading: Grade 1 = mild, no intervention needed; Grade 2 = Moderate, local or non-invasive intervention needed; Grade 3 = Severe, not immediately life-threatening, may need hospitalization, Grade 4 = Life threatening with urgent intervention indicated, Grade 5 = Death. Arachnoiditis not defined until CTCAE 5 = inflammation of arachnoid membrane adjacent to SAS |
| Additional: trauma | Cord involvement; motor vehicle accident; projectile or gunshot wound; industrial accident, time from injury to arcs; syrinx development; response to surgical intervention |
| Additional: brain | Specific cranial nerve involved, response to surgery, response to other treatment, arcs found at surgery, cause |
| Additional: diagnostic tests | test parameters; association of test result to clinical symptoms, differential diagnosis (in free text) |
| Additional: genetic | Family tree, gene identified |
| Additional: rheumatologic | Diagnosis: therapy used; response of CNS disease in addition to systemic |

(*Continued*)

**Table 4.** (Continued)

| Type of data element | Data captured |
|---|---|
| Additional: spinal and brain surgery | Fusion vs non-fusion; time from surgery to new symptoms; dose of rhBMP-2; other devices used; legal publications about devices in free text; post-surgical follow up; time to first MRI; use of steroids; time from surgery to arcs |
| Additional: neuraxial anesthesia | Cleansing agent; anesthetic; # attempts to enter epidural space or IT space; return of blood; accidental puncture of dura; blood patches used; time from event to arcs symptoms; course of arcs symptoms, post-discharge follow-up |
| Additional: myelogram dye | Dye used; time to arcs; # myelograms done; time from dye injection to arcs; steroids used; dye drained |
| Additional: spine pain injections | Skin cleansing; Procedure; # fluoroscopy used; # steroid used; type anesthetic used; time from procedure to arcs symptoms; |
| Additional: subarachnoid hemorrhage | Extent; treatment; peripheral fibrinolytic markers levels as predictors of severe residual; attempted intrathecal therapy and response, complications |

surrogate to divide countries into more (OECD members) and less-economically developed (non-member of OECD, N-OECD) [18]. We collected specialty-specific data, also shown in Table 4. Oncology was the only area standard study data required replacement with different metrics.

## Synthesis methods and effect measures

A qualitative synthesis was performed on data from all 887 studies, shown in Table 5. Given heterogeneity of results dependent on cause, treatment, or imaging diagnosis, data were first grouped by categories and subcategories, with weighted means and totals calculated. Data cleaning and table calculations were conducted using R (R Core Team, 2019). Occurrence or reoccurrence of DLMs and baseline characteristics are included in this dataset.

Difference in weighted means was used as the effect measure for categorized data. Fig 3 shows distribution of articles and patients per article per year. P values were used to calculate the impact of removal of biased or incorrect data sources. Removal of studies generating a statistically significant impact on results (p≤0.05) were independently scrutinized by all authors, requiring unanimous agreement.

## Bias determination and maintaining study integrity

This is the consolidated PRISMA 2020 bias section. Every source of data underwent study bias assessment, extraction, and data check by ≥2 independent reviewers using systematic review software designed to blind researchers to each other's responses until the final "disagreement resolution" phase. Disagreements at any part of the PSR were settled by modified Delphi processes and author vote.

## Bias and critical appraisal for study inclusion

Table 3 contains the questions and results for bias, performed on each study. As part of critical appraisal, we verified study data calculation and coding. For studies with data mistakes or questionable results, investigators were contacted. Of eight questionable sources, none were included. Six authors did not reply. General biases such as short follow-up and incomplete neurologic status reporting were seen in nearly all studies and did not result in exclusion.

**Table 5. Full PSR sources.** This table contains all data charted for the study and used for synthesis, with references. Data are sorted into major category, then subcategory. (NOTE: The reference numbers are specific to this table and do not correspond to article reference numbers). The year span of papers within a subcategory is included to help the reader understand that changes in therapy over time may have occurred and as a gauge as to when LM damage from diseases were first identified.

| Category | Subcategory (+Reference) | | # sources | N pt. | Total Male | Mean Age | Total Arcs | Tot LM | N-OECD | OECD | Y min | Y max | study types |
|---|---|---|---|---|---|---|---|---|---|---|---|---|---|
| Autopsy/Pathology Total [19–22] | | | 4 | 139 | 61 | 21.2 | 9 | 15 | 0 | 4 | 1921 | 2017 | 4-CS |
| Cause | Chiari [23–27] | | 5 | 214 | 138 | 44.8 | 31 | 31 | 1 | 4 | 1983 | 2018 | 5-CS |
| | Chronic Meningitis [28–51] | | 23 | 616 | 296 | 47.0 | 35 | 97 | 1 | 22 | 1828 | 2019 | 19-CS | 4-OS |
| | Congenital Or Perinatal [52–59] | | 8 | 123 | 54 | 3.1 | 101 | 101 | 1 | 7 | 1985 | 2019 | 6-CS | 2-OS |
| | | | | | | | | | | | | | |
| | Genetic [60–64] | | 5 | 18 | 6 | 42.1 | 15 | 18 | 0 | 5 | 1973 | 2017 | 5-CS |
| | Igg4 [65] | | 1 | 10 | 4 | 53.4 | 0 | 1 | 0 | 1 | 2010 | 2010 | 1-CS |
| | Infection | Amoeba [66,67] | 2 | 4 | 3 | 5.7 | 1 | 3 | 1 | 1 | 1982 | 2001 | 1-CR | 1-CS |
| | | Aseptic Arcs [68–71] | 4 | 6 | 2 | 45.8 | 4 | 6 | 0 | 4 | 1980 | 2015 | 2-CR | 2-CS |
| | | Listeria [72,73] | 2 | 3 | 3 | 59.0 | 2 | 3 | 0 | 2 | 1995 | 2003 | 1-CR | 1-CS |
| | | Typical Bacteria [74–79] | 6 | 313 | 173 | 45.4 | 12 | 12 | 1 | 5 | 1976 | 2018 | 4-CR | 1-CS | 1-RCT |
| | | TOTAL Bacterial Meningitis | 8 | 316 | 176 | 45.5 | 14 | 15 | 1 | 7 | 1976 | 2018 | 5-CR | 2-CS | 1-RCT |
| | | Fungus | Aspergillus [80–82] | 3 | 3 | 3 | 30.0 | 2 | 3 | 0 | 3 | 1980 | 2015 | 2-CR | 2-CS |
| | | | Blastomycosis [83] | 1 | 1 | 1 | 17.0 | 1 | 1 | 0 | 1 | 1973 | 2019 | 9-CR | 1-CS | 2-OS | 1-MCRS |
| | | | Coccidiomycosis [84–90] | 7 | 215 | 170 | 40.3 | 68 | 103 | 0 | 7 | 1983 | 2013 | 3-CR |
| | | | Cryptococcus [91–95] | 5 | 10 | 7 | 41.0 | 8 | 9 | 1 | 4 | 1990 | 1973 | 1-CR |
| | | | Exserohilum Rostratum [96] | 1 | 440 | 183 | 65.0 | 223 | 223 | 0 | 1 | 2020 | 2018 | 1-CR | 4-CS | 2-OS |
| | | | Histoplasmosis [97,98] | 2 | 2 | 1 | 31.5 | 1 | 2 | 0 | 2 | 2012 | 2017 | 4-CR | 1-CS |
| | | | Other Fungi [99,100] | 2 | 2 | 1 | 19.5 | 2 | 2 | 0 | 2 | 1950 | 2020 | 1-OS |
| | | TOTAL Fungal Meningitis | 20 | 667 | 361 | 58.3 | 299 | 337 | 1 | 19 | 1950 | 2020 | 2-CR |
| | | Bannwarth's Syndrome [101–104] | 4 | 17 | 10 | 50.9 | 5 | 16 | 0 | 4 | 1982 | 2015 | 2-CR |
| | | EVB, VZV, Other Common Viruses [105–117] | 13 | 339 | 39 | 37.9 | 2 | 27 | 0 | 13 | 1990 | 2019 | 13-CR | 4-CS | 3-OS |
| | | HIV-AIDS-HTLV3 [118–125] | 8 | 470 | 240 | 37.4 | 84 | 268 | 2 | 6 | 1985 | 2014 | 2-CR | 2-CS | 4 -OS |
| | | HTLV-1 [126] | 1 | 24 | 7 | 46.2 | 3 | 3 | 1 | 0 | 1990 | 1990 | 1-CCS |
| | | Lyme-Neuroborreliosis [127–140] | 14 | 1105 | 660 | 47.5 | 4 | 475 | 0 | 14 | 1984 | 2019 | 5-CR | 5-CS | 4 -OS |
| | | NCC = Neurocysticercosis [141–165] | 25 | 2181 | 1390 | 32.3 | 273 | 414 | 16 | 9 | 1980 | 2019 | 6-CR | 7-CS | 11 -OS | 1-CCS |
| | | Neurobrucellosis [166–170] | 5 | 34 | 23 | 35.2 | 3 | 30 | 5 | 0 | 2011 | 2018 | 3-CR | 2 -OS |
| | | Other Infections [171–175] | 5 | 2626 | 31 | 47.5 | 4 | 4 | 3 | 1 | 1973 | 2017 | 1CS | 3 -OS | 1-MCRS |
| | | Other Tick Disease [176–178] | 3 | 1356 | 201 | 48.0 | 0 | 963 | 0 | 3 | 1996 | 2014 | 1CR | 2 -OS |
| | | Syphilis [136,179–184] | 7 | 21 | 15 | 44.6 | 18 | 18 | 0 | 7 | 1937 | 2016 | 3-CR | 4-CS | 1-RCT |
| | | TB [119,185–259] | 76 | 4139 | 1984 | 29.7 | 655 | 1711 | 57 | 19 | 1951 | 2020 | 35-CR | 19-CS | 24 -OS | 1-CCS | 1-AS |
| | | Toxoplasma [260] | 1 | 1 | 0 | 28.0 | 1 | 1 | 1 | 0 | 2001 | 2001 | 1-CR |
| | | West Nile Virus [261–264] | 4 | 20 | 2 | 50.5 | 4 | 4 | 0 | 4 | 2003 | 2015 | 4-CR |
| | | Worms And Other Parasites [265–271] | 7 | 46 | 30 | 33.6 | 3 | 30 | 5 | 2 | 1988 | 2018 | 5-CR | 2-CS | 1-RCT |
| | | Zika [272,273] | 2 | 25 | 13 | 45.3 | 3 | 25 | 1 | 1 | 2018 | 2019 | 1CS | 1-CCS |
| | | TOTAL INFECTIONS | 207 | 13180 | 5097 | 39.4 | 1316 | 4286 | 93 | 113 | 1937 | 2020 | 96-CR | 53-CS | 50-OS | 4-CCS | 2-RCT | 1-AS | 1-MCRS |
| | IT Injection: Non-Cancer | IT Injections Non Cancer [68,69,274–290] | 19 | 43 | 23 | 45.6 | 28 | 31 | 1 | 18 | 1954 | 2019 | 15-CR | 4-CS | 4-OS |
| | Cancer | Leukemia/Lymphoma· [291–315] | 24 | 8208 | 4563 | 13.2 | 76 | 104 | 1 | 23 | 1973 | 2018 | 3-CR | 10-CS | 3-OS | 3-RCT | 2-P2 | 3IG |
| | | Metastatic To LM [316–332] | 14 | 408 | 176 | 47.1 | 111 | 298 | 3 | 11 | 1996 | 2019 | 5-CR | 2-CS | 3-OS | 3-RCT | 1-P1 | |
| | | Primary CNS [333–349] | 16 | 182 | 84 | 13.7 | 27 | 70 | 1 | 15 | 1990 | 2014 | 7-CR | 4-CS | 1-OS | 2-RCT | 1-P1 | 1PS | |
| | | TOTAL Leptomeningeal Cancer | 54 | 8798 | 4823 | 18.7 | 214 | 472 | 5 | 49 | 1973 | 2019 | 15-CR | 16-CS | 7-OS | 8-RCT | 2-P1 | 2-P2 | 1PS | 3IG |
| | Myelogram Dye [5,70,350–384] | | 37 | 5212 | 312 | 44.4 | 1163 | 1174 | 4 | 33 | 1928 | 2017 | 17-CR | 6-CS | 13-OS | 1-P1 |
| | Neural Cysts [385,386] | | 2 | 9 | 3 | 34.8 | 2 | 2 | 0 | 2 | 1970 | 1995 | 1-CR | 1-CS |
| | Neurotoxicity Of Immunotherapy [387–389] | | 3 | 19 | 14 | 62.3 | 16 | 17 | 0 | 3 | 2016 | 2019 | 1-CR | 1-CS | 1-OS |
| | Post-LP [390–392] | | 3 | 3 | 2 | 44.7 | 3 | 3 | 1 | 2 | 1978 | 2013 | 3-CR |
| | Regional Anesthesia [393–431] | | 39 | 73303 | 3377 | 45.2 | 92 | 183 | 7 | 32 | 1945 | 2019 | 17-CR | 13-CS | 9-OS |
| | SAH [163,425,432–462] | | 32 | 189 | 56 | 49.2 | 29 | 50 | 3 | 29 | 1922 | 2019 | 19-CR | 12-CS | 1-OS |
| | Sarcoid [463–470] | | 8 | 89 | 42 | 43.7 | 6 | 49 | 0 | 8 | 1978 | 2018 | 4-CR | 2-CS | 2-OS |
| | SLE, RA, Other Autoimmune [471–488] | | 16 | 82 | 64 | 49.6 | 16 | 27 | 4 | 12 | 1968 | 2019 | 10-CR | 6-CS |
| | Spinal Abnormality [489] | | 1 | 5 | 5 | 48.8 | 5 | 5 | 0 | 1 | 1972 | 1972 | 1-CS |
| | Spinal Epidural Steroid Injection Or Pain Injection [490–500] | | 11 | 10859 | 272 | 50.7 | 35 | 123 | 0 | 11 | 1967 | 2019 | 3-CR | 5-CS | 2-OS | 1-RCT |
| | Spine Devices [501–504] | | 4 | 127 | 54 | 50.7 | 4 | 4 | 0 | 4 | 1991 | 2018 | 2-CR | 1-CS | 1-P2 |
| | Surgery-Fusions | Fusion Surgery No BMP [13,502,505–510] | 8 | 1427 | 519 | 51.8 | 10 | 85 | 2 | 6 | 2000 | 2018 | 2-CS | 5-OS | 1-MA |
| | | hrBMP [13,15,510–519] | 13 | 7897 | 1421 | 51.1 | 1 | 1153 | 0 | 13 | 2009 | 2019 | 4-CS | 5-OS | 1AE | 1DBS | 1 MA | 1 MDB |
| | | TOTAL Surgery-Fusions | 19 | 9324 | 1940 | 51.4 | 11 | 1238 | 2 | 17 | 2000 | 2019 | 6-CS | 9-OS | 1-AE | 1-DBS | 1-MA | 1-MDB |
| | Surgery Non-Fusion [503,504,520–547] | | 30 | 1395 | 791 | 32.4 | 231 | 515 | 6 | 24 | 1947 | 2018 | 9-CR | 17-CS | 2-OS | 1-P2 | 1PRNR |
| | Trauma [548–559] | | 12 | 174 | 94 | 38.8 | 72 | 88 | 5 | 7 | 1945 | 2016 | 5-CR | 4-CS | 2-OS | 1-CCS |
| Cause Total | | | 531 | 123660 | 17411 | 43.4 | 3418 | 8506 | 133 | 397 | 1828 | 2020 | 233-CR | 161-CS | 104-OS | 5-CCS | 10-RCT | 3-P1 | 3-P2 | 1-PRNR | 1-AE | 1-DBS | 1-PS | 3-IG | 1-AS | 2-MCRS | 1-MA | 1-MDB |

*(Continued)*

**Table 5.** (Continued)

| Category | Subcategory (+Reference) | # sources | N pt. | Total Male | Mean Age | Total Arcs | Tot LM | N-OECD | OECD | Y min | Y max | study types |
|---|---|---|---|---|---|---|---|---|---|---|---|---|
| **Cranial And Cranial Nerve Involvement** | Acoustic [560–564] | 5 | 429 | 6 | 46.5 | 12 | 12 | 1 | 4 | 1972 | 2015 | 4-CR \| 1-OS |
| | Cerebellar [327,565–574] | 11 | 158 | 80 | 30.6 | 157 | 157 | 4 | 7 | 1924 | 2018 | 5-CR \| 6-CS |
| | Cerebral [575] | 1 | 153 | 0 | | 153 | 153 | 0 | 1 | 1966 | 1966 | 1-CS |
| | Optic Chiasm [93,180–182,184,231,241,242,259,387,559,576–614] | 50 | 469 | 226 | 36.4 | 280 | 280 | 11 | 39 | 1930 | 2019 | 28-CR \| 17-CS \| 4-OS \| 1-CCS |
| | Trigeminal [615–630] | 16 | 973 | 430 | 55.9 | 213 | 213 | 8 | 8 | 1988 | 2019 | 2-CR \| 11-CS \| 2-OS \| 1-CCS |
| **Cranial And CN Total** | | 83 | 2182 | 742 | 49.6 | 815 | 815 | 24 | 59 | 1924 | 2019 | 35-CR \| 39-CS \| 7-OS \| 2-CCS |
| **Diagnosis And Imaging** | CT [631–636] | 6 | 221 | 56 | 57.7 | 52 | 52 | 0 | 6 | 1976 | 1987 | 4-CS \| 2-OS |
| | EMG [637,638] | 2 | 12 | 1 | 61.0 | 2 | 2 | 0 | 2 | 1976 | 1986 | 1-CS \| 1-OS |
| | Endoscopic Diagnosis [97,639–641] | 4 | 107 | 50 | 56.2 | 69 | 69 | 0 | 4 | 1993 | 2012 | 1-CR \| 2-CS \| 1-OS |
| | General X-Rays [287,642] | 2 | 12 | 8 | 40.3 | 9 | 9 | 1 | 1 | 1936 | 2013 | 2-CS |
| | LM Biopsy [215] | 1 | 115 | 53 | 59.7 | 3 | 3 | 1 | 0 | 2017 | 2017 | 1-CS |
| | MRI [276,412,475,549,643–657] | 42 | 2251 | 911 | 47.9 | 201 | 250 | 9 | 33 | 1985 | 2020 | 9-CR \| 24-CS \| 8-OS \| 1-CCS |
| | Myelogram [350,373,658–666] | 11 | 473 | 255 | 46.5 | 172 | 172 | 0 | 11 | 1928 | 2015 | 2-CR \| 7-CS \| 1-OS \| 1-P1 |
| | Nerve Root Infiltration For Diagnosis [665] | 1 | 62 | 0 | | 12 | 12 | 0 | 1 | 1988 | 1988 | 1-CS |
| | Next Generation Sequencing [28,667] | 2 | 2 | 1 | 27.5 | 1 | 1 | 0 | 2 | 2014 | 2019 | 2-CR |
| **Diagnosis And Imaging Total** | | 71 | 3255 | 1335 | 48.9 | 521 | 570 | 11 | 60 | 1928 | 2020 | 14-CR \| 42-CS \| 13-OS \| 1-CCS \| 1-P1 |
| **Differential Diagnosis Total** [336,342,348,668–684] | | 20 | 336 | 24 | 30.8 | 38 | 45 | 4 | 16 | 1927 | 2014 | 10-CR \| 10-CS |
| **Failed Back Surgery Syndrome Total** [650,653,685] | | 3 | 99 | 47 | 41.6 | 30 | 30 | 2 | 1 | 1980 | 2017 | 2-CS \| 1-P2 |
| **Unclassifiable Arachnoiditis Total** [686–698] | | 12 | 377 | 230 | 35.7 | 364 | 368 | 3 | 9 | 1937 | 1991 | 3-CR \| 7-CS \| 1-OS \| 1-RCT |
| **Outcomes And Complications** | Arachnoid Cysts [443,699–706] | 8 | 169 | 61 | 51.9 | 30 | 30 | 4 | 4 | 1996 | 2018 | 5-CR \| 3-CS |
| | Arachnoid Fibrosis [550,707–716] | 11 | 206 | 109 | 44.8 | 136 | 136 | 2 | 9 | 1957 | 2016 | 4-CR \| 4-CS \| 2OS \| 1-CCS |
| | Autonomic Disturbances [717] | 1 | 1 | 1 | 44.0 | 1 | 1 | 1 | 0 | 2001 | 2001 | 1-CR |
| | Bowel/Bladder Incontinence, Sexual Dysfunction [718] | 1 | 1 | 0 | 13.0 | 1 | 1 | 1 | 0 | 1971 | 1971 | 1-CR |
| | Cauda Equina Syndrome [719] | 1 | 1 | 1 | 60.0 | 1 | 1 | 0 | 1 | 1985 | 1985 | 1-CR |
| | Chronic Inflammation [720] | 1 | 26 | 6 | 47.9 | 26 | 26 | 0 | 1 | 2017 | 2017 | 1-CS |
| | ICH Hydrocephalus [721,722] | 2 | 71 | 0 | 19.5 | 3 | 27 | 1 | 1 | 1995 | 2015 | 1-CR \| 1-CS |
| | Neuropathic Pain [723] | 1 | 8 | 5 | 44.6 | 8 | 8 | 0 | 1 | 1980 | 1980 | 1-CS |
| | Ossificans [282,290,552,580,656,724–763] | 45 | 65 | 31 | 49.5 | 63 | 64 | 15 | 30 | 1930 | 2019 | 36-CR \| 9-CS |
| | Other Complications [764–766] | 3 | 4 | 1 | 55.3 | 4 | 4 | 0 | 3 | 1990 | 2017 | 2-CR \| 1-CS |
| | Progression [767–769] | 3 | 6 | 3 | 19.5 | 6 | 6 | 0 | 3 | 1951 | 2006 | 2-CR \| 1-CS |
| | Syringomyelia [21,60,72,195,203,206,217,219,227,281,439,443,447,551,558,573,588,655,707,714,731,732,736,751,759,770–821] | 77 | 1679 | 726 | 37.7 | 771 | 1029 | 14 | 63 | 1940 | 2019 | 36-CR \| 36-CS \| 5-OS |
| **Outcomes And Complications Total** | | 146 | 2229 | 941 | 39.2 | 1042 | 1325 | 35 | 111 | 1930 | 2019 | 81-CR \| 57-CS \| 7-OS \| 1-CCS |
| **Treatment** | Bladder Stimulation [822] | 1 | 1 | 0 | 50.0 | 1 | 1 | 0 | 1 | 2008 | 2008 | 1-CR |
| | Cordotomy/Rhizotomy [588,715,823–827] | 7 | 209 | 103 | 44.8 | 35 | 69 | 0 | 7 | 1960 | 2010 | 6-CS \| 1-OS |
| | CSF Shunt [828–840] | 13 | 270 | 56 | 27.3 | 175 | 187 | 2 | 11 | 1971 | 2019 | 4-CR \| 8-CS \| 1FU |
| | Intraspinal Drugs [69,497,498,841–844] | 7 | 263 | 63 | 46.3 | 63 | 63 | 1 | 6 | 1984 | 2015 | 1-CR \| 5-CS \| 1-OS |
| | Lysis Of Adhesions [845–849] | 5 | 169 | 100 | 48.9 | 47 | 47 | 2 | 3 | 1978 | 2019 | 1-CR \| 2-CS \| 2-OS |
| | Other Drugs [850] | 1 | 6 | 3 | 48.7 | 2 | 2 | 1 | 0 | 2011 | 2011 | 1-CS |
| | Radiation [851] | 1 | 17 | 0 | 42.2 | 17 | 17 | 0 | 1 | 1962 | 1962 | 1-CS |
| | Rehab [724,852,853] | 3 | 53 | 34 | 44.4 | 9 | 9 | 1 | 2 | 1995 | 2018 | 1-CR \| 1-CS \| 1-OS |
| | Spinal Cord Stimulation [275,710,811,844,854–859] | 10 | 165 | 71 | 48.1 | 76 | 126 | 1 | 9 | 1974 | 2016 | 4-CR \| 4-CS \| 2-OS \| 1-RCT |
| | Spinal Injections [685] | 1 | 26 | 0 | | 26 | 26 | 0 | 1 | 1980 | 1980 | 1-CS |
| | Surgical Management [219,570,574,670,752,770,810,860–892] | 40 | 1089 | 699 | 41.5 | 791 | 925 | 6 | 33 | 1924 | 2017 | 14-CR \| 17-CS \| 9-OS |
| **Treatment Total** | | 88 | 2278 | 1092 | 42.2 | 1242 | 1472 | 14 | 73 | 1924 | 2019 | 26-CR \| 45-CS \| 16-OS \| 1FU |
| **Major Category Total: This Row Includes 70 Duplicated Files That Informed More Than One Category** | | 958 | 134572 | 21881 | 43.4 | 7474 | 13142 | 226 | 730 | 1828 | 2020 | |
| **Total for the SR, no duplications** | | 887 | 133261 | 21321 | 43.5 | 7153 | 12721 | 209 | 676 | 1828 | 2020 | 370-CR \| 337-CS \| 142-OS \| 1-FU \| 8-CCS \| 11-RCT \| 3-P1 \| 3-P2 \| 1-PNRS \| 1-DBQ \| 1-PS \| 1-AE \| 3-IGS \| 1-AS \| 2-MCRS \| 1-MA \| 1-MDB |

Several sources (71/887 = 8%) appear in ≥2 categories if the paper made significant contributions to several categories. As an example, a randomized trial of spine fusion ± rhBMP-2 with serial MRI would appear in 3 categories: spine fusion with rhBMP, spine fusion without rhBMP and MRI.

Abbreviations: N-OECD = Non-member Organization for Economic Cooperation & Development and OECD = Organization for Economic Cooperation & Development Member; CR = case report; CS = case series; OS = observational study; RCT = randomized clinical trial; CCS = case-control study; MCRS = multi-center retrospective study; P1 = Phase 1; P2 = Phase 2; INS = internet survey; DBQ = data base query; PS = pilot study; IGS = intergroup study; AS = ambispective; PNRS = prospective non-randomized study; FU = follow-up to previously reported study, Study type AE = Adverse events reported to public FDA MAUDE database and published; MA = meta-analysis; MDB = Medicare database.

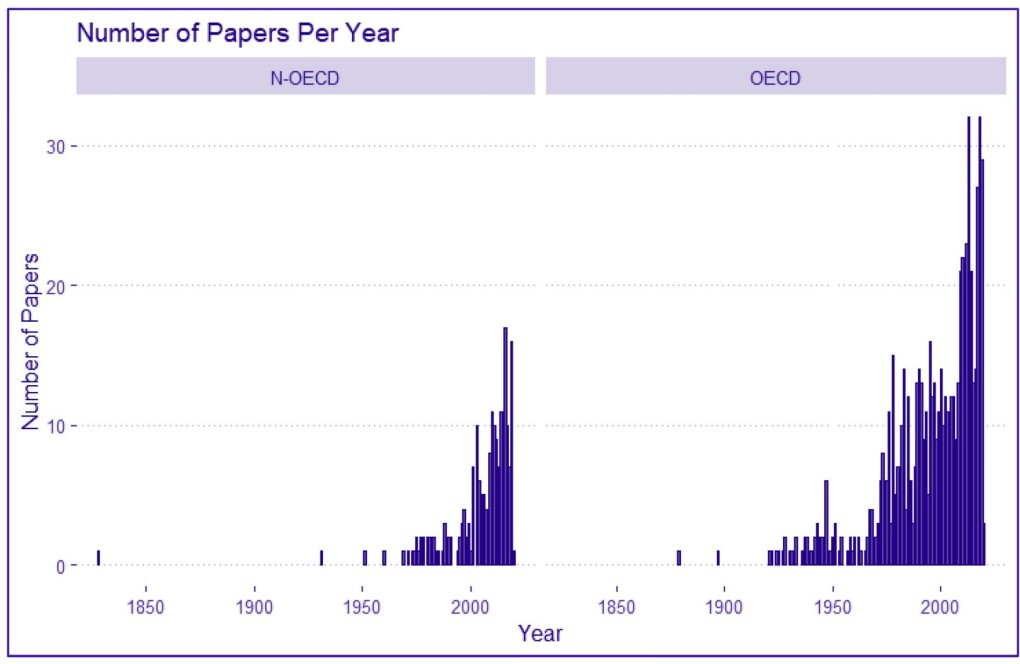

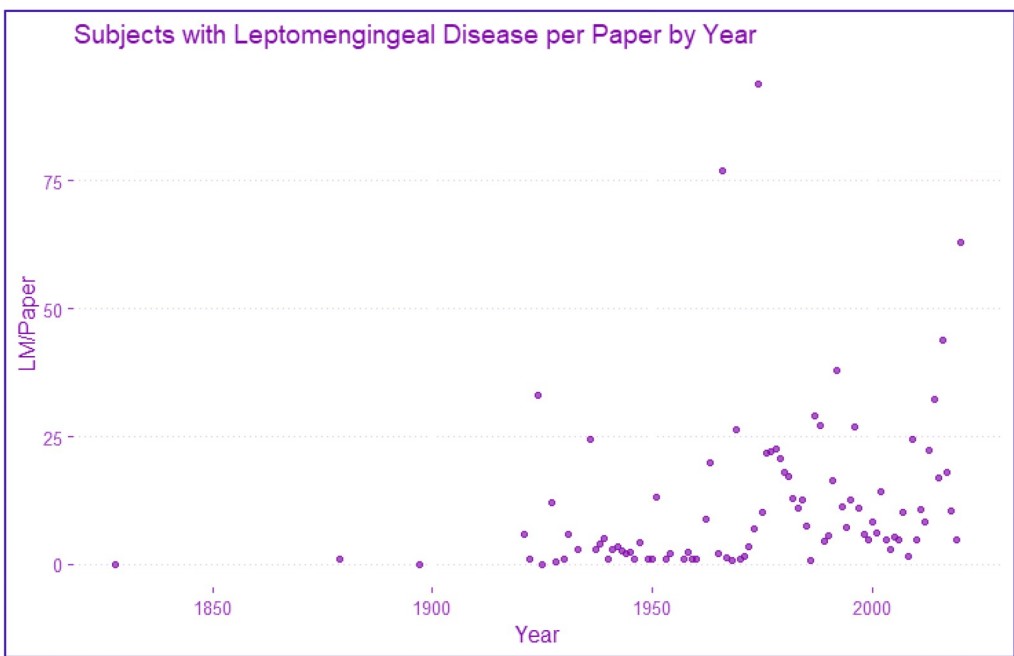

**Fig 3. The bar-chart displays the distribution of publishing year for papers in this PSR stratified by N-OECD versus OECD.** In both regions the number of papers published has increased over time indicating that research in the area is intensifying. The scatter plot displays the number of subjects diagnosed with DLMs per paper over time. The increase in variation over time exhibits the change from mostly case reports to larger scale observational studies and randomized controlled trials.

Two heavily utilized treatments among patients in FB groups were excluded after websites were reviewed and physicians contacted. The first involved 2018 versions of frequently changed cocktails of "neuro-steroids and neurohormones" called "Tennant Protocols" [893]. While we obtained a list of older animal studies no objective results in human patients were

provided for analysis. The second involves commercial use of neural stem cells (NSC) with reported dramatic resolution of arachnoiditis symptoms in patients. No response was received to request for data [894].

*Five studies excluded due to quality or bias deserve mention as they might be included in attempts to repeat this PSR without doing critical review.* Each study had a statistically significant impact on results, four with $p < 0.0001$ and one with a $p = 0.01$. Every author independently appraised each article. Three large spinal fusion ± rhBMP-2 studies with thousands of patients each were excluded from analysis—one with unexplained data errors after author discussion [895], one missing arachnoiditis codes [896], and one considered financially biased with unexplained doubled AE in the control group [897]. Two other studies were excluded due to serious methodology errors [898,899].

## Risk of bias in included studies

As shown in Table 5, studies ranged from 1828–2021 (due to e-publication ahead of print), without time-period breakdown. During this span of 200 years, multiple advances were made across all areas of medicine. A study limited to the past 20 years would better represent real world data today.

## Bias from lack of reporting in studies

No assumptions were made. Missing information was recorded as absent. Mean follow-up < 12 months was common in surgery patients and <1 month for pain injections and neuraxial anesthesia (NAA). This was particularly problematic in studies using surgical lysis of adhesions as attempted *treatment* for LM fibrosis(LMF). Both positive and negative neurologic findings at "reported study points" were rarely complete. Two large anesthesiology studies were dependent on voluntary reporting and returned surveys, with bias likely. Sexual dysfunction was regularly reported for males but never for females.

## Bias in our results due to handling of missing data

Many spinal surgery studies were funded by manufacturers of a study component, leading to disincentives to report negative outcomes. When results were missing, we counted them as zero, absent, not improved, or not performed. No assumptions that unreported data were normal were made. Few studies addressed patient sense of quality of life (QOL) with their disabilities or permanent sequelae, or whether pain was adequately palliated.

## Methodology certainty assessment

We have high certainty in the body of evidence due to: (1) the experience of the author team; (2) rigorous adherence to ≥2-person methodology; (3) access to macroscopic, microscopic, and molecular pathology; (4) an abstract review team composed of multiple nurse-patients trained in systematic review in addition to the modified e-Delphi team; (5) a modified e-Delphi team composed of three academic neurosurgeons, a practicing neurologist, a PhD-level molecular geneticist/patient, a PhD-level biostatistician/caregiver, multiple methodologists, and six additional physician/patients.

## Results and discussion

### Studies included

Baseline characteristics of each included source as well as references are in Table 5. The PRISMA diagram (Fig 1) shows 14774 studies screened, with 6% (887) included. Case reports

accounted for 41.9% (372/887) of sources, 0.3% (372/133261) of total patients, and 2.9% (372/12721) of DLM patients. Most reports (59.9%) involved causation, not including cranial and cranial nerve cases, accounting for 66.9% (8506/12721) DLMs patients. Non-database sources in Fig 1 contain of 127 hand-retrieved studies, and gray sources. Fig 3 shows trends in individual studies. In both N-EOCD and EOCD, number of papers and patients per paper have increased yearly, reflecting more and larger studies with time.

## Key aspects of meningeal formation and anatomy

The brain and spinal cord arise from neural tube ectoderm. The meninges are of mesenchymal origin. The dura has no tight junctions between cells and is not the CSF-blood/body barrier. Chemicals, inflammatory chemokines, biologics, and many small substances easily pass through the dura [900,901]. The arachnoid and pia maters together make up the leptomenix (singular) or leptomeninges (LM) when plural. Fig 4A shows the location of the arachnoid barrier layer (arachnoid membrane) in the cauda equina [902]. The nerve roots within the cauda equina cistern are covered only by pia, *therefore enhancement of the covering of swollen nerve roots on MRI represents acute pial inflammation (or leptomeningitis) but not arachnoiditis.*

The inner arachnoid layer is composed of different arachnoid fibroblastic-type cells which form the *arachnoid trabeculae to pia structures that create the SAS, the open framework through which the pial perineuronal vascular plexus, nerves, and CSF pass* (Fig 4B and 4C at the spinal cord level). The SAS contains macrophages and the NSC niches. NSC migrate throughout life into the adjacent CNS to replace injured neurons removed by autophagy [903]. Destruction of the LM with loss of local NSC means the anterior and posterior horns of the spinal cord and brainstem, as well as the neurons near the surface of the brain die from spinal cord injury (SCI), without replacement. The pia encases blood vessels and nerves until just before the nerves exit the foramen, where the arachnoid and pia join [901,904]. Fig 4C is a scanning electron micrograph of the SAS of a rat brain showing the differences between the arachnoid barrier layer (a), arachnoid trabecula (*) and pia mater (c) [905].

Substances in or around the dura or paraspinal area can diffuse into the spinal blood vessels prior to entry into the LM. They enter the SAS via receptors or diffusion as shown in Fig 4D. Alternatively, substances can be directly injected into the SAS or into the ventricles where they will eventually be distributed into the SAS. The illustration in Fig 4D shows the role of the glymphatics, CSF, and vasculature [906,907]. The glymphatics are part of the NVU and contribute to nutrient provision and waste removal critical to sustaining CNS neuron life [906]. All figures are used with written permission from original creators.

A question important to the DLMs patient communities has been, "are spinal DLMs considered a SCI?" Patients report the medical community shows more compassion towards patients diagnosed with a SCI. No patient wants to have to explain arachnoiditis or be coded as having "chronic pain," after which care is tainted with the underlying question of drug seeking behavior. There is clinical, autopsy, and pathology proof areas of the brain and spinal cord with inadequate collateral blood supply die when blood vessels and nerves traversing the SAS are destroyed, as detailed in recent neuroscience reviews [900,906–909]. Patients with DLMs and upper neuron localization have resultant SCI. However, CSN diseases can occur in the absence of DLMs.

## The term "arachnoiditis" missed 43% of dlms patients

Table 2 shows 42/63 (66.7%) terms for DLMs did not containing the term arachnoiditis. As shown in Table 5, "arachnoiditis" alone or combined with another term yielded 7153 patients, while the full search for DLMs returned 12721 patients. Thus 43.4% (5518/12721) of DLMs

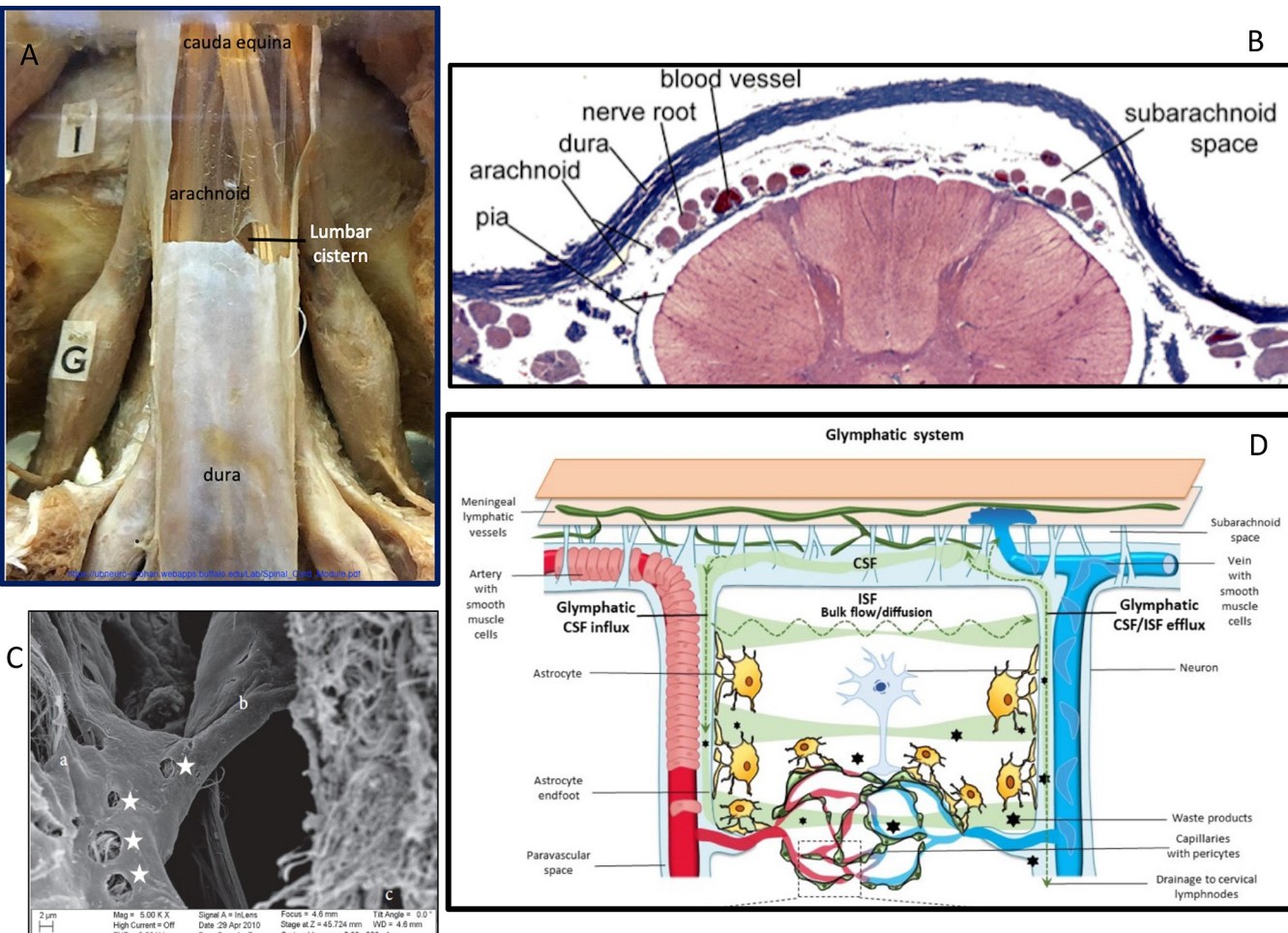

**Fig 4. Visualizing the LM from macro to microscopic. (A)** shows a human autopsy specimen at the level of the cauda equina, demonstrating the arachnoid closely following the dura. The nerve roots are wrapped in the pia membrane.† **(B)** is a cross-section of a mouse spinal cord with surrounding meninges, demonstrating how many critical structures traverse the SAS.† **(C)** is a SEM of an arachnoid trabecula {b*} traversing from arachnoid barrier layer {a} to pia {c} showing the porous nature of the trabecula.† **(D)** The glymphatic schematic and BBB demonstrating the critical role of unimpaired CSF flow into the CNS from and back to the SAS.§ Pictures: † Courtesy of C. Cohen PhD, Curator of The Brain Museum, University of Buffalo NY, used with written permission. § Micrograph and illustration ©F. Fornai MD (31) used with written permission.

patients would have been missed without our rigorous strategy. Only 62/5588 (1.1%) cases not using arachnoiditis were diagnosed exclusively as chronic meningitis. A landmark paper from *1943* had 11 different terms for LMDs presented and stated "arachnoiditis" was incorrect terminology, had "Arachnoiditis (Diffuse Proliferative Leptomeningitis)" as the title [19].

Table 6B (column 5) shows iatrogenic DLMs from spinal fusion surgery are called radiculitis 99.1% of the time, as are 71.5% of spinal injection DLMs, and 55.1% of spinal non-fusion surgery DLMs. Non-procedural causes with well-ingrained names and significant incidences include: subarachnoid hemorrhage (SAH), TBM, NB (from various forms of Lyme), neurocysticercosis (NCC), neurobrucellosis, neurosarcoidosis, "neurologic syndromes" from arthropod-transmitted diseases (arboviruses such as Chikungunya and Zika) [272], chemical meningitis, carcinomatous meningitis, and less common terms.

Regardless of names, autopsy [19–22] and innumerable microscopic reports contained in the PSR are clear and consistent. The areas involved are the SAS + pia components of the LM,

**Table 6. Characteristics of major categories by economic status (A), and use of "arachnoiditis" terminology in subcategories of DLMs (B): All sources were divided into major categories, then further divided into subcategories.** Part A shows numbers of papers, total patients, and numbers from N-OECD and OECD. Part B shows papers, sources, those who had diagnoses of DLMs (without the term "arachnoiditis"), and % of total population in the specific subcategory who had diagnoses of DLMs (inclusive of arachnoiditis). Of 887 sources, 2 had undeclared location of patients, removed from economic columns for clarity.

| A. Primary Category | category | N papers | N patients | # N-OEDC | # OECD | N-OECD:OECD patients |
|---|---|---|---|---|---|---|
| **All Sources: Major categories** | Autopsy & Pathology | 4 | 139 | 0 | 139 | 0:139 |
| | **Cause (* in Section B = subcategories of causes)** | **531** | **123660** | **13730** | **107347** | **1:7.8** |
| | Cranial & cranial nerve | 83 | 2182 | 683 | 1499 | 1:2.2 |
| | Diagnosis and Imaging | 71 | 3255 | 1017 | 2238 | 1:2.2 |
| | Differential Diagnosis | 20 | 336 | 288 | 48 | 6:1 |
| | Failed Back Surgery Syndrome | 3 | 99 | 73 | 26 | 2.8:1 |
| | Unclassifiable arcs | 12 | 377 | 185 | 192 | 1:1 |
| | Outcomes & complications | 146 | 2228 | 155 | 274 | 1:1.8 |
| | Treatment | 88 | 2278 | 272 | 1973 | 1:7.3 |
| | Total sources in all categories (some duplicated) | 958 | 134572 | 16403 | 115459 | – |
| | Total quantitative source N, (no duplications) | 887 | 133261 | 16043 | 114609 | 1:7.1 |
| **B. Category or Subcategory 1** | **Selected subcategory 2** | **N sources** | **N patients** | **N (%) + DLMs w/o arachnoiditis terminology** | **n (%) DLMs in specific patient population** | **% DLMs of total CATEGORY** |
| *Genetic: ADD | NA | 5 | 18 | 3 (16.7) | 18 (100) | 0.2 |
| *Chronic Meningitis | NA | 23 | 616 | 62 (63.9) | 97 (15.7) | 1.1 |
| *SAH | NA | 32 | 162 | 21 (42.0) | 122 (73.5) | 1.4 |
| *Communicable diseases | **Total for CD in all causes** | **207** | **13180** | **2970 (69.3)** | **4286 (32.5)** | **50.4** |
| | Bacterial ± spine involvement | 8 | 316 | 1 (6.7) | 15 (4.7) | 0.2 |
| | Exserohilum rostratum (2020 follow-up) | 1 | 440 | 0 | 223 (50.7) | 2.6 |
| | Chronic fungal meningitis | 20 | 667 | 38 (11.3) | 337 (50.5) | 4.0 |
| | Lyme NB | 14 | 1105 | 471 (99.2) | 475 (43) | 5.6 |
| | NCC neurocysticercosis (2020 follow-up) | 25 | 2181 | 141 (31.4) | 414 (19) | 4.9 |
| | TBM | 76 | 4139 | 1056 (61.7) | 1711 (41.3) | 13.1 |
| | EVB, VZV, Other Common Viruses | 13 | 339 | 25 (92.6) | 27 (8.0) | 0.3 |
| | Neurobrucellosis | 5 | 34 | 27 (9.0) | 4 (20) | 0 |
| | Zika | 2 | 25 | 22 (88.0) | 25 (100) | 0.3 |
| | Other tick-borne diseases with CNS involvement | 3 | 1356 | 963 (100) | 963 (71) | 11.3 |
| *Spinal surgeries | corrective spine surgery, non-fusion | 30 | 1395 | 284 (55.1) | 515 (36.9) | 6.1 |
| | **Spine surgery fusion NO hrBMP** | **8** | **1427** | **75 (88.2)** | **85 (6.0)** | **1.0** |
| | **Spine surgery fusion + hrBMP** | **13** | **7897** | **1152 (99.9)** | **1153 (14.6)** | **13.6** |
| *Non-surgery spine interventions | Myelogram Dye | 37 | 5212 | 11 (0.9) | 1174 (22.5) | 13.8 |
| | Neuraxial Anesthesia | 39 | 73303 | 91 (49.7) | 183 (0.2) | 2.2 |
| | Spinal injections | 11 | 10879 | 88 (71.5) | 123 (1.1) | 1.4 |
| *Trauma | NA | 12 | 174 | 16 (18.2) | 88 (50.6) | 1.0 |
| *Cancer Related | Neurotoxicity of immunotherapy | 3 | 19 | 1 (5.9) | 17 (89.3) | 0.2 |
| | Chemotherapy given into CSF for leukemia and lymphoma prophylaxis | 24 | 8208 | 28 (26.9) | 104 (1.3) | 1.2 |
| *Rheumatologic | Sarcoid | 8 | 89 | 43 (87.8) | 49 (55.1) | 0.6 |
| | RA, SLE, AS | 16 | 82 | 11 (40.7) | 27 (32.9) | 0.3 |

(*Continued*)

**Table 6.** (Continued)

| | | | | | | |
|---|---|---|---|---|---|---|
| **Cranial and Cranial Nerve** | TOTAL | 83 | 2182 | 0 (0) | 815 (37.4) | 100 |
| | Optic Chiasm | 50 | 469 | 0 (0) | 280 (59.7) | 34.4 |
| | Acoustic | 5 | 429 | 0 (0) | 12 (2.8) | 1.5 |
| | Trigeminal | 16 | 973 | 0 (0) | 213 (21.9) | 26.1 |
| **FBSS** | NA | 3 | 99 | 62 (63.9) | 30 (30.3) | 100 |
| **Complications** | syringomyelia | 77 | 1679 | 258 (25.1) | 1029 (61.3) | 77.7 |
| | LM fibrosis | 11 | 206 | 0 | 136 (66.0) | 10.3 |
| | Progression | 3 | 6 | 0 | 6 (100) | 0.5 |
| **Treatment** | TOTAL | 88 | 2278 | 230 (15.6) | 1472(64.6) | 100 |
| | Surgical lysis of adhesions from LM fibrosis | 40 | 1089 | 134 (14.5) | 925 (84.9) | 62.8 |
| | Cordotomy/Rhizotomy | 7 | 209 | 34 (49.3) | 69 (33.0) | 4.7 |
| | Shunt | 13 | 270 | 12 (6.4) | 187 (69.3) | 12.7 |
| | Intraspinal drugs | 7 | 263 | 0 | 63 (24.0) | 4.3 |
| | Rehabilitation | 3 | 53 | 0 | 2 (3.8) | 0.1 |
| | Spinal Cord Stimulator | 10 | 175 | 50 (39.7) | 126 (72) | 8.6 |
| | Spinal Injections | 1 | 26 | 0 | 26 (100) | 1.8 |
| **Total in study inclusive of entire population** | | **887** | **133261** | **5568 (43.8)** | **12721 (9.5)** | **NA** |

DLMs going by names other than arachnoiditis are found in nearly every area of medicine, including rare autosomal dominant genetic transmission with 100% penetration causing early, severe DLMs [60,61].

not the outer arachnoid layer or membrane. In humans, clinical symptoms of acute inflammation may lead to symptoms of LMF if the inflammatory to fibrotic cascades are not stopped.

## DLMs and associated symptoms are not rare

Using 2020 USA and global population numbers, and disease incidence (prior to SARS-CoV2), we calculate >3.7M global and >311.8K cases in the USA annually. The annual incidence of DLMs for the limited etiologies in Table 7, globally and in the USA, are 94x and 190x the "rare disease" designation of 0.5:100K, respectively. Global numbers are artificially low as unavailable data is counted as zero.

Results indicate three "basic areas" of similar symptomatology. **Table** 8 contains the most common symptoms described in the 12721 patients. Dozens of case reports document the diagnosis of DLMs up to a decades after the presumed inciting event.

In brain meningitis, the high incidence of upper and lower extremity radiculitis might be related to occult spinal disease, but this is an interesting finding indicting the need to fully image the neuroaxis in worsening patients. Cerebral and base of brain leptomeningitis involves the spine in 80% of patients, particularly in TB and HIV+ patients [185]. In the majority of cases, MRI of the spine is not performed, despite neurological symptoms relating to the spine. Even in *cured* bacterial leptomeningitis 20–80% will have permanent neurologic or cognitive difficulties [920,924].

DLMs are common, have many causes as shown in Tables 5 and 8, and every physician should learn to recognize the symptoms then perform a thorough history to determine whether MRI and lumbar puncture (LP) are needed to rule out curable causes in the chronic setting. The use of next generation sequencing (NGS), where available, can make a diagnosis in the face of negative cultures, serology, and pathology [28,667].

**Table 7. Calculated real world incidences of LMDs in selected diseases based on WHO and CDC incidences.** [Δ] Data is based on minimum incidences of DLMs noted in the PSR* or published X 2020 population numbers. The incidence of DLMs is far greater than the "rare disease" threshold of 1:200K or 39,500 annually in the US.

| Cause | Specific population calculated | Incidence of LMDs used (* = from this SR) | Total number procedures, if needed for calculation | US yearly LMD cases | Global yearly LMD cases |
|---|---|---|---|---|---|
| Neuraxial anesthesia | OB[910] 3747540 births, 79% epidurals | 0.2%* | 2960557 epidurals US [910] Unable to find Global N** | 5,921 | Not reported |
| | Surgery [408] -incidence higher in teaching hospital and high comorbidities | 25.7:100k epidural abscess + hematoma vascular surgery in US 4.2:100K in UK | 1382805[911] (US) 707455 (UK) Use UK/G population for global | 9,133 | 331,800 |
| Spine surgery | Only spinal fusion included | LMD is 6% no rhBMP-2* & 14.6% +rhBMP-2* in our study, if 14.7% use rhBMP-2 for fusion, [912] total population incidence is 7.2% | -2015 Global 3.5 M spine fusions [913] -2018 US fusions: 589566 (455435 IP, 30040 refusion, 103981 ASC) [914,915] | 42,449 | 252,000 |
| Epidural pain injections | | * 1.1% AE | Total 2017123:US population [916] Unable to find global N* | 20,171 | Not reported |
| Total iatrogenic | | | | 77,674 | 583,800 |
| Trauma and Road Accidents | Trauma-w/SCI | 17810[917] 500000 [918] | | 17,810 | 500,000 |
| SAH [919] | | 7.9:100K Global 6.9:100K US | | 30,000 | 624,100 |
| Non-CDs | | | | 47,810 | 1,124,100 |
| TBM [920] | Tubercular meningitis | 1–500,000 | | 84 | 250,000 |
| NCC [921] | neurocysticercosis | 0.23/100000 | | 1,000 | 18,170 |
| Lyme disease[921] | neuroborreliosis | 206/100k | 16,274,000/16%NB 2603840/20%LMD US = 1/3 cases 300k NB | 100,000 | 520,768 |
| Cryptococcal meningitis [922] | | 220,000 globally [923] 41.3/100k US 1/3 die | In HIV not on HAART | 84,527 | 223,100 |
| Bacterial meningitis [920,921] | "WHO DEFEAT BY 2030" goal, (target vaccine preventable) | Africa incidence 5M – 300K death/year per WHO US inc. 4100–500 death/y | 20% survivors have neurologic deficits (DLMs) is used to calc total cases | 720 | 940,000 |
| Total selected CDs above | | | | 186,331 | 1,952,038 |
| **TOTAL CASES of LMDs /YEAR (limited to above causes); **unable to find global incidences of ESI and spinal anesthesia** | | | | **311,815** | **3,659,938** |
| **Minimum Incidence (includes only causes above)** | | | | **>0.10% 95:100K** | **>0.05% 47:100k** |

Rare disease threshold 1:200K = 39,500 globally | US population = 328.2M | 2020 global population = $7.9 \times 10^9$.

[Δ] These numbers are not corrected for halted elective surgery due to Covid.

## Non-iatrogenic causes of DLMs

**Communicable diseases.** ***The most common cause of DLMs in the PSR was CDs***, tabulated in the final column of Table 6B, accounting for 50.4% of patients in the "causes" category. This result surprised us, as we assumed iatrogenic causes would predominate, as stated on all reputable USA websites in 2018. WHO and CDC yearly incidence reports were used for Table 7, demonstrating in 2021, CDs make up almost roughly 53% of global and 60% of USA cases of chronic DLMs. The global data is skewed by missing cases of iatrogenic causes in non-US OECDs. The rapidly rising incidence of tick-borne diseases contributes to both columns. The full range of CDs identified are in Table 5, with infectious causes spanning from 1828–2021. Many diseases included are now routinely cured or controlled in OECDs. Studies demonstrated in >70% of acute leptomeningitis, the spine was also involved, although it is rarely

**Table 8. Common symptoms per area involved occur.** Symptoms were remarkable similar across the 3 areas presented (spine, base of brain, brain). Spine and CN DLMs produce the same types of symptoms but localized to the area involved during early phases. If available, response to CDs therapy is included.

| Spine symptoms from DLMs | CN and base of brain symptoms from DLMs | Brain symptoms from DLMs |
|---|---|---|
| Neuropathic pain predominates in almost all patients: described as burning, electric shocks, and bizarre skin sensations. Pain often impacts multiple dermatomes. 40% disabled due to pain alone | Radicular and neuropathic pain, not limited to face most common finding | Headache |
| Sometimes bimodal pattern of initial spinal symptoms, temporary recovery, then different symptoms as below | 25% radicular face pain resistant to morphine plus NSAIDs | pseudotumor cerebri (HA, papilledema) |
| Bowel and bladder incontinence is an early sign in most patients with lower spine involvement | CN palsy, (facial nerve involved first with NB) | meningismus |
| Approximately 1/3 of patients complain of ambulation difficulty due to weakness or balance dysfunction | HA | Fogginess |
| Sensory numbness, hyperesthesia, and loss of vibratory sense more common than motor weakness, | Oculomotor nerve palsy | Hallucinations |
| Motor weakness | Meningitis symptoms | High incidence of associated thrombotic stroke |
| Neurogenic claudication, with back pain occurring after walking or performing an activity for a period is a hallmark | Myelitis with spasticity of or weakness in legs | Cognitive dysfunction |
| myoclonus | dizziness | Delirium |
| Movement-associated muscle cramps and fasciculations | normal bowel and bladder function in all patients if lower spine not involved | Slurred speech to aphasia |
| Radicular pain also occurs, related to area involved | Syringomyelia as a long-term AE | Hydrocephalus, may require shunt |
| Syringomyelia as a long-term AE | If damage is severe enough to cause death to anterior and/or posterior horn cells, lower symptoms may occur | Fetal death (anencephaly or microcephaly) from some acute causes (Zika) while pregnant |
| Arachnoid cysts and webs as AE | MRI presence LM fibrosis with no surgical management predicts lack of improvement | Ataxia |
| Muscle spasticity as a later sign | new therapies being tried in children with TBM and acute vision loss (thalidomide and infliximab) | muscle tremors |
| | Hydrocephalus-may require shunt | normal bowel and bladder in all patients |
| | IF cause is infectious and treatment begun within a week of symptoms, 85% respond with decrease in symptoms BUT 59% have residual symptoms after infection cured | associated radiculitis in 30–40% |
| | IF cause is infectious and treatment delayed > 30 d, 56% of neurologic symptoms get better with treatment BUT most have persistent radicular pain despite microbiologic cure | neuropsychiatric symptoms common |
| | | acute bacterial meningitis in the US: 30% die. Of those cured 20% have above residuals |

imaged. The USA Center for Disease Control (CDC) reports even in fully cured acute meningococcal meningitis, long term disability occurs in 20% of patients [924].

Not included in the study data but worth mentioning is the virus SARS-CoV-2 is neurotrophic with neurologic symptoms as a common manifestation and LM involvement documented [925–927]. One review demonstrates the virus infects tissue via the ACE2 receptor, also expressed on vascular endothelial cells that allows infected lymphocytes to enter the SAS. The virus is reported to move up nerves, crossing synapses, and entering the CNS via a "Trojan Horse" mechanism [928]. Neuroimaging, brain autopsy, and LPs demonstrate spinal cord and LM involvement are consistent with neurologic residuals in recovered patients [929–931].

**Subarachnoid hemorrhage.** Of 32 sources on SAH synthesized, the death rate was 13.6–25%. Rate of LMF was 75.3% (122/162) of survivors. Most patients who survived developed LMF symptoms 1–10 months post SAH and stabilized or improved over time. The PSR incidence of SAH was only 1.4% of all causes. However, using current data, Table 7 shows incidence rates for SAH, with 30K and >624K cases per year in the USA and globally, respectively [919], making SAH from ruptured aneurysm the second most common of DLMs globally. In the USA, SAH is third, after spinal fusion surgery.

Free blood in the SAS initiates a complex reaction of neuroinflammation, collagen deposition, and arachnoid fibrosis. Multiple factors are involved, beyond the scope of this review [932]. The CSF has low-to-absent plasminogen and high levels of plasminogen activator inhibitors leading to inefficient fibrinolysis of blood clots [933]. The end result is LMF, with significant disability and NPP.

**Trauma from road accidents causing spinal cord injury (SCI).** The PSR returned 12 sources with 88 patients affirming most traumatic SCIs also involve LM injury. Time span was 1945–2016. We speculate the 50% incidence is low due to unavailability of MRI for much of the time frame. Disruption of spinal cord blood flow that enters through SAS vasculature is responsible for further demise after initial injury [934]. Table 7 shows globally, trauma from road injury is third most common cause of irreversible DLMs and resultant spinal cord ischemia globally. The rate is especially high in areas where transportation safety requirements are substandard due to poor restraints and road conditions. Data from hundreds of sources demonstrate improvement in spinal cord function from NSC. We reviewed the actual pictures in these sources. Spinal cord architecture and function improved in many NSC trials, but none demonstrate regeneration of the LM, which are critical to attain or maintain normal function after SCI.

**Treatment or prophylaxis for LM cancer.** Review of oncology intra-CNS therapy data revealed the need to align neurological toxicity with consistent descriptions of DLMs symptoms. In the past, "arachnoiditis" has been separated from nerve changes, which is now known to be contrary to the disease process. Per Table 5, oncology data was further divided into primary CNS cancers, metastatic solid tumors, and leukemia or lymphoma trials. This division was an attempt to assess long term neurologic consequences in patients getting attempted curative therapy, generally childhood hematologic malignancy patients.

The 54 trials had 8798 patients, with leukemia constituting 93.3% (8208/8798), spanning 1973–2019. The toxicity grading system is different than any other field and changes every few years. Determining the incidence of acute DLMs is impossible from past publications and would require chart or EMR review using updated neurologic toxicity definitions.

In hematologic malignancies intrathecal (IT) therapy is often used prophylactically to prevent CNS relapse in children with curable malignancies. There were 104 instances (1.3.%) of long-term neurologic deficits after CNS prophylactic therapy, but oncology-specific neurologic criteria were used. Standardized criteria could yield different results. Route of administration may be an issue for further research. The same dose is given intraventricularly via Ommaya reservoirs as is used IT via LP, despite differences in CSF volume of distribution and cellular make-up of the 2 compartments. We consulted with two renowned hematologic malignancy experts, Richard Larson MD (email July 23, 2021) and Dieter Hoelzer MD PhD (email April 6, 2021) with both concurring little is known about the clinical impact of differences in routes of administration for therapeutic or prophylactic IT treatment. There is still work to be done in this area.

**Neurotoxicity of cancer immunomodulators.** Rare but potentially life-threatening toxicities associated with immune-modulating drugs used in cancer treatment occur, including neurotoxicity. The 3 studies of neurotoxicity from checkpoint inhibition and/or ipilimumab,

demonstrated 89% (17/19) of the neurologic symptoms were due to DLMs. Considering the widespread use of these drugs, this is an important issue. For immune-related AE, oncology guidelines are available to assist with when to stop treatment as well as when to use steroids, infliximab, or other anti-interleukin therapies [935].

## Iatrogenic causes or associations

**Early myelogram dye indisputably caused diffuse LMF.** Results from the myelogram dye studies in Table 5 represent a fraction of the global suffering that resulted from diffuse LMF caused by SAS injection of non-resorbable oil-based and early water-based myelogram dye. Law suits were settled with government health authorities in multiple countries, with details on incidence not in medical literature [936]. Multiple myelogram agents caused diffuse, permanent "adhesive arachnoiditis" or "arachnoiditis ossificans" in addition to iophendylate (Pantopaque in the USA, Myodil in other countries), including Dimer-X, Kontrast U, Conray 60, Thorotrast, and others.

There were 37 studies with 5212 patients using different dyes. In total, 22.5% (1174/5212) cases of severe, progressive LMF were *reported*. In the USA, iophendylate was FDA approved under the 510K device law. Iophendylate myelogram dye was not withdrawn from the US market until 1987, years after initial publications on DLMs. We found no reported cases of DLMs with the current myelogram dye, iohexol.

**DLMs from spinal fusion surgery.** There were 515/1395 DLM cases in 30 reports (36.9%) of corrective non-fusion spine surgery, with multiple procedures per patient. Most had baseline congenital, traumatic, neoplastic, or previous scoliosis correction surgery with extensive hardware revisions, as well as high numbers of pre-surgery LMF. No articles discussed the impact of symptoms of DLMs on quality of life. The high incidence of DLMs in patients with multiple spine surgeries and revisions has significant implications for diagnosis and treatment. Recurrent symptoms may not be surgically correctable but related to LMF. As MRI techniques and radiologist recognition improve, such patients found to have DLMs are ideal candidates for clinical trials.

Elective spinal fusion surgery using rhBMP-2 (recombinant human bone morphogenetic protein-2) has been controversial. Tables 7B and 8 show of included studies, the weighted means of occurrence of DLMs is 6.0% in the 1427 patients without rhBMP-2 and 14.6% in the 7897 patients receiving rhBMP-2 (p<0.0001). In most cases of lumbar surgery, the rhBMP-2 was used off-label. There are other FDA approved biologics on the market which may not have sufficient clinical safety and efficacy data to know whether their use may be associated with DLM. Ongoing studies utilizing lower doses of rhBMP-2 may determine if adverse events are dose-dependent.

**Neuraxial anesthesia.** We identified 39 sources with 186 patients who developed severe neurologic complications and LMF after NAA "misadventure." Blood patches used for dural leaks accidentally injected into the SAS can also cause LMF [937]. The incidence of LMF from post-obstetrical NAA was 0.2% (182/73303). The two largest trials were dependent upon voluntary physician reporting of AE. Speculation of cause ranged from skin cleaning solutions to anesthetic concentration. Incidence of complications appears related to the level of individual experience and volume of procedures done, with high volumes associated with better results. References for Table 7 also indicate DLMs are higher for patients with obesity, more comorbidities, procedures performed in teaching or rural hospitals, and in non-obstetrical vascular surgery.

Interestingly, 2/3 of patients developing LMF post NAA complained about neurologic symptoms at the time of needle placement [393]. A recent anesthesia review states pre-existing

neuropathy places patients at higher risk for immediate complications and permanent new neurologic deficits post NAA [938]. We recommend not performing blind NAA in patients with DLMs. In a patient with advanced lumbosacral DLMs, blood vessels and nerves may not be mobile, thus are unable to move out of the way of the needle.

**Spinal pain injections.** The PSR incidence of DLM after epidural steroid and transforaminal injections is 1.1% (123/10879) in 10 sources, although long term follow up is inadequate. Reports range from transient paralysis and acute DLM to permanent cauda equina syndrome. A 2021 systematic review of transforaminal injections, not included in our data, discusses but does not quantify resultant LMF [939]. In 2016 major insurers required a trial of ESI prior to approving lumbar spine surgery. This is no longer the case as of January 2022 for several companies.

**FBSS and additional surgical procedures in those with DLMs.** There were 3 studies with 99 patients addressing whether FBSS represents DLMs. DLMs occurred in 30% (30/99). However, 70% had surgically correctable lesions. These patients should be evaluated at centers with radiologists, spine surgeons, and patient advocates expert in dealing with DLMs if a surgically correctable cause is also suspected. We found no evidence that adjacent-level operative procedures or LP in *non-DLM involved* areas worsens existing disease.

**N-EOCD vs OECD incidences.** Table 6A divides results by number of sources, patients, and economic distribution. Using N-OECD:OECD ratio, sources discussing causation of DLMs where overwhelmingly from OECD countries with a ratio of 1:7.1. Total data sources in the PSR also favored publication from OECD at 1:3 (Table 5). Many Injuries, deaths, and diseases in N-OECD often are unreported in English-translatable literature searches. This leads to disparities and inaccuracies. In this PSR, N-OECD data predominated results for NCC, neurobrucellosis, TBM, and other worms and parasites. All of these are curable diseases with chemotherapy and adequate water sanitation.

In Table 7 we estimate real-world contemporary incidences using current incidence rates with CDC and WHO published disease and total population information. When available, we used the most current incidence data, as cited, otherwise we used the incidence data calculated from our study, shown with an * for identification. Minimum incidence of DLMs, in the limited diseases included, demonstrate rates of >0.10% (95:100K) and >0.05 (47:100K) in the USA and globally, respectively. The global numbers are falsely low as we used zero (0) when unable to find global (including non-USA OECD) data. Epidurals for childbirth are routine in other OECD, but almost never used in N-OECD due to lack of trained anesthesiologists. A similar situation exists for epidural and transforaminal pain injections.

Other disparities impacting high N-OECD rates of DALY due to DLMs include previously discussed right rates of SCI from automobile accidents, lack of HAART for HIV+ individuals, high rates of death from acute infectious meningitis due to lack of basic antibiotics, and now high rates of Covid deaths due to lack of availability of vaccines. Chronic DLMs with permanent sequelae are far from rare both in the US and globally, with an incidence of roughly 200 and 100x the rare disease standard of 0.5:100K. Additional analysis of PSR data by economic status will be presented separately.

## Diagnostic studies for DLMs

**Pre-MRI imaging studies.** For the PSR, all imaging studies were included. There were 71 studies containing 3255 patients spanning from 1928–2020. The earliest study reported was the use of iodized oils to image "lesions of the spinal cord" from the research and translational lab producing them in Stockholm [350]. The earliest article on imaging and suggested treatment of arachnoiditis via air injection was published in 1942 [658]. The procedure was used to

treat arachnoid cystic lesions blocking CSF flow (1/6 cases responded and 3/3 attempted lami-nectomies had no response). Studies published in the 1960's-70 reported oil and early water-based dyes, all causing high incidences of diffuse LMF. The first CT of the lumbar spine was in 1976 [631], but it was the first MRI in 1985 that significantly changed CNS and spinal cord imaging [940]. CT demonstrates osseous structures of the spine better than MRI and is still used.

**MRI with contrast.** MRI techniques continue to advance, and ≥1.5T MRI with contrast is the current test of choice for DLMs. Recent small studies confirm the frequently quoted 1990 Delamarter arachnoiditis staging is outdated [643]. Instead arachnoid cysts, nerve root clumping, enhancement or displacement are more reliable [644,941].

A 2021 study looks at interrater reliability (IRR) in classifying 1.5 and 3T MRIs in 96 patients with known DLMs between three neuroradiologists and two musculoskeletal radiolo-gists. IRR was very poor for staging or synechiae, however in the "most experienced" readers, was strong for synechiae [942]. One study suggesting MRI in prone position is helpful to detect clumping and adhesion since normal cauda equina nerves tend to follow gravity when free-floating [943]. A newer technology, constructive interference in steady-state (CISS) and fast imaging employing steady-state acquisition with phase cycling (FIESTa–c) has excellent visu-alization of CSF flow and DLMs, but a high learning curve to interpret correctly [944]. When the cause of radicular pain is in question, we recommend high quality MRI with contrast in the prone position when looking for DLMs. Currently MRI is indispensable for diagnosis of DLMs but is only as good as the experience ability of the reader.

Not included in the PSR but presented for completeness is an systematic review of MRI LM enhancement (LME) of the brain in multiple sclerosis and other neuroinflammatory diseases e-published prior to print January 2022 [945]. DLMs were not specifically included. A large differential of inflammatory diseases of the LM did have MRI LME consistent with clinical symptoms. Normal controls showed 6/163 (3.7%) LME. The occurrence of LME in "normal controls" has been used by a by two physicians claiming arachnoiditis or DLMs are not a real entity [686,946]. Our results and conclusions remain unimpacted. MRI can detect DLMs in the inflammatory stage when there is a possibility of preventing progression to LMF. Radiolo-gists should be provided with as much history as possible to encourage attention to the area of interest.

**Performing a diagnostic LP when DLMs is known or suspected.** We found multiple studies where no known precipitating factor was present (prior brain or spine procedure, trauma, meningitis, IT chemotherapy or craniospinal axis radiation). Since actionable results can be obtained from CSF sampling in patients without prior procedures, we recommend LP if peripheral exam and work up does not yield a diagnosis. In two patients with chronic fever and symptoms of chronic leptomeningitis, but multiple negative LPs, diagnosis was made by LP with next generation sequencing (NGS) in both, which is increasingly available and useful [28,667]. Blind endoscopic leptomeningeal biopsy was found to be ineffective pre-MRI, as shown by multiple studies (Table 5). A 2017 N-OECD report of open MRI-guided LM biopsy yielded TBM (NGS was not available).

The PSR demonstrated multiple etiologies can cause LMF. When acute infection is not a concern, after thorough exam and studies looking for a systemic cause of neurologic disease, we recommend a contrast MRI of the lumbosacral spine (as well as brain if symptomatic or immunocompromised). The reason for the spine MRI is to determine if there is LMF and to what extent. If the MRI demonstrates "peripheralization" or "empty sac" of the cauda equina nerve roots, consider cisternal CSF sampling. With LMF and empty sac, pia-enclosed vascular bundles, and nerves of the cauda equina are adherent to the dural sac, so they cannot move out of the way of a needle as would normally occur. Putting a needle through this area could result

in catastrophic worsening of disease. Preparation should be made ahead of time to obtain all samples needed, including cultures, gram stain, cytology, immune studies, pathology, and NGS if needed. No reports of successful ultrasound guided in LMF without vascular trauma were found. The PSR yielded 3 reports of arachnoiditis caused by traumatic LP [390–392].

**Suspected acute CD leptomeningitis in a patient with known LMF.** This PSR specifically excluded acute leptomeningitis. Acute infectious leptomeningitis is a medical emergency, however LP in a patient with *known* lumbar LMF can lead to catastrophic complications, as in the above section. The spinal cord ends in most people at T12-L1. If the location of the LMF is shown to be localized to L4/5 or lower, there is a chance of cauda equina mobility at the L2 or L3 interspace. If previous diagnostic MRI is not available for review, we recommend cisternal puncture for CSF sampling if the patient does not have an implanted CSF reservoir (Ommaya).

**Disease course and prognosis.** This PSR yielded 2718 patients with permanent spinal DLMs from large studies (myelogram dye, spine fusion surgery, neuraxial anesthesia, and spinal injections). Many case reports and smaller case series increase this number to >3000. Infectious causes added another >4000 cases. Adding in base of the brain, of the nearly 8600 cases, the striking similarity of area per area symptoms of spinal DLMs allows us to report common features and adverse events.

**Acute DLMs can be fatal.** Acute CD leptomeningitis, brain or brainstem SAH, acute hydrocephalus from CSF flow blockage, head or high cervical spinal trauma can all be fatal. Fatalities from acute CD leptomeningitis are far higher in N-OECD than OECD. A "Defeat by 2030" WHO goal is elimination of vaccine-preventable leptomeningitis deaths as shown in Table 7. We purposefully excluded acute CD-related leptomeningitis from this PSR due to lack of follow-up in the current system of inpatient-only hospitalists.

The only study on "lifespan" was excluded for major methodology flaws. There is no evidence non-progressive LMF *per se* reduces survival, but epidemiology studies show patients with mobility limitations have a higher risk for SDOH that contribute to earlier death, such as obesity, Type 2 diabetes, and cardiac deconditioning [947]. Non-opioid medications used for NPP may be associated with weight gain. Walking distances can be difficult because of neurogenic claudication. Other exercises should be substituted. Depression is common after developing a serious chronic illness. Teaching coping skills and using multidisciplinary support teams is important, particularly for intractable NPP patients.

**Most patients do not have a continually progressive course.** We found 6 patients in 3 reports documenting progression, with 2/6 occurring after NAA, and 4/6 years after neonatal meningitis. Both NAA patients also had oil-based myelograms as part of their post-NAA injury work-up. Two of 4 of the post meningitis cases occurred after attempted NAA made them dramatically worse [767–769]. In all 6, progression occurred after a second SAS insult.

In the chronic setting, everything progressive or toxic can be fatal, as can suicide due to lack of palliation for those with intractable chronic NPP. *Outside of these situations, we found no evidence chronic DLMs continually progress in the absence of continual pathologic stimulation.* However, CNS neuron death due to many causes can occur. With no repair mechanism, those cells will die and cause symptom progression.

For iatrogenic cases, the PSR spinal surgery data generated 49 studies with >10K patients, with all articles claiming the majority who developed a new acute post-operative leptomeningitis (current term radiculitis) resolved by 3–6 months (Table 5). Given the biphasic nature of some cases of LMF, no statements about overall progression to permanent DLMs can be made. Adequate follow up for these studies is lacking.

The population-based NB study in Denmark yielded 431 documented cases of DLM, 16 also with encephalitis. Only 14 (3.2%) had no treatment response. When treatment ended, 101

(28.1%) had residual symptoms, with 52 (51.5) being radicular pain. There was a significant incidence difference between those who delayed treatment >30 days (28.2% in delayed vs 4.9% p<0.001), suggesting there is a period during which acute leptomeningitis can be prevented from becoming permanent LMF. Steroids were not used in the study [127]. Long term data from cured childhood brain tumor studies indicate early neurologic deficits are irreversible.

**Outcomes and complications.**   Stopping fibrotic transformation, blockage of CSF flow, and subsequent consequences should be a high priority target for future research. Without development of new treatments, those who develop LMF are unlikely to resolve, and clinical course will depend on amount of CSF flow blockage. Table 8 lists typical characteristics brain, brainstem and CN, and spine involvement. Disability from CDs, SAH, trauma, and iatrogenic causes appear in $\geq 2$ of the top 10 disabilities in every country we examined in the WHO list of DALY [10,948].

**Paradoxical reaction to TBM treatment.**   There were 8 studies with 163 patients describing a paradoxical reaction to TBM treatment. This important manifestation was first recognized in AIDS patients not on HAART presenting with TBM. Reconstitution of the immune response increases or causes new symptoms due to inflammatory LM reactions, despite concurrent TBM treatment. The most concerning consequences are the possibility of vision loss, severe hydrocephalus, and acute spinal cord compression. HIV treatment for the untreated HIV+ patient is not started until weeks after TB treatment is initiated. It also occurs in HIV negative malnourished children and adults, a common occurrence in N-OECD. Managed appropriately, it does not adversely impact outcome [186]. Recently, the impending vision loss from a paradoxical reaction in a 7-year-old has been successfully managed with infliximab [187,188].

**Flares.**   All patients who join the dedicated FB groups learn about "flares," a common manifestation of DLMs and one of the greatest causes of distress to patients. The PSR yielded no specific article on flares, although several detailed early case reports described the typical painful manifestations post-provocation occurring days later. During a flare, symptoms get worse for days to weeks, then return to baseline. Patients report flares to be triggered by overactivity, prolonged sitting, atmospheric pressure changes and airplane travel.

Flares also cause anxiety and fear about progression. One 2022 arachnoiditis review, not included in the PSR, discusses "flare-ups." [949] Every member of the PPI has flares, requiring escalation of treatment for a short time, at times with no known provocations as noted above. They are a frequent cause for ER visits for parenteral steroids and repeat MRIs. No data on best approaches were identified.

**Fasciculations and muscle spasms occur in LMF.**   Fasciculations and uncontrolled muscle spasms occur in patients with spinal involvement [950–953]. Daily episodes involving radicular units respond well to diazepam and stretching. Additional recommendations from the American Association of Neurological Surgeons include (1) removable casting or bracing, though many patients find this cumbersome; (2) muscle relaxants such as baclofen, cyclobenzaprine, methocarbamol, or tizanidine; (3) botulinum toxin injections; (4) IT baclofen or selective dorsal rhizotomy for recalcitrant spasticity [954]. Many patients have worsening episodes at night, finding foot flexor to knee braces uncomfortable. Mayo Clinic spinal injury rehabilitation unit recommends placing pillows between the footboard and feet to keep feet flexed during sleep, as well as staying well hydrated (author consultation).

## Sexual activity is impacted by severe DLMS, stressing relationships

This was a measure we attempted to capture since relationship support is foundational to dealing with severe chronic disease. Several reports in Table 5 discussed decreased sexual

performance or impotence in men upon diagnosis. No source reported female sexual function. There are other contributing factors, such as depression, which is common in patients with serious and complex illnesses. Drugs used to treat either NPP or depression have known side effects on sexual function.

DLM patients have painful muscle spasms that occur with certain positions commonly used during sexual activity. The PSR revealed a single case report of a 60 y/o man who developed repeated episodes of painful priapism requiring surgical decompression of the penis followed by myelogram demonstrating a complete block at L3-4 and laminectomy demonstrating central LMF of the cauda equina, with nerves stuck together. Post operatively he did well with normal sexual function [719].

After SCI, unless the sacral nerves are involved, physiologically >50% can achieve male ejaculation or female orgasm [955]. Sacral nerve injury S2 and below drastically decreases chances of orgasm in both males and females to 0–17%. Autonomic dysreflexia can occur with injuries above T7 [956–958]. Bowel and bladder sphincter disturbances pose additional challenges. Resources can assist practitioners with communication of techniques and sex toys if uncomfortable with this topic [955].

## SAS cysts and webs, hydrocephalus and syringomyelia are proven sequala of DLMs

There were 146 papers with 59.4% (1325/2229) patients developing serious permanent complications of LMF with CSF block. Eight articles demonstrated arachnoid cysts and webs in 30/169 (17.8%) patients, all also having syrinxes. Arachnoid webs appear to be remnants of arachnoid cysts. Hydrocephalus is commonly reported with both acute and chronic DLMs with 58/877 (6.6%) sources reporting feasibility and outcomes of various types of shunts for hydrocephalus as the primary intervention. Of 11 with LMF in the title, 136/206 (66%) had extensive pathology sampling demonstrating abolished SAS architecture causing CSF block.

The PSR yielded 77 sources with 1029/1679 (66.6%) patients developing syringomyelia due to LMF with CSF block. Patients with symptomatic syringomyelia underwent neurosurgical management, with most recurring within the first 10 years, depending on degree of lysis of arachnoid adhesions performed.

SAS cysts and webs that disrupt flow, when fenestrated, can provide temporary relief from some symptoms for many years [959]. If new neurologic changes occur after the first years of stable LMF, patients should be evaluated for progression to arachnoid cysts, syringomyelia, or hydrocephalus.

## Is chronic cauda equina syndrome (CES) from DLMs ± flare considered a surgical emergency?

There are still multiple patient and physician resources that state **all** CES is a surgical emergency. Acute or sub-acute **compressive** CES is a surgical emergency. With the recognition of DLMs, much literature will have to be revised appropriately [960–963]. Chronic CES from iatrogenic LMF is very distressing to patients, particularly those moving in and out of bowel and bladder sphincter control with disease fluctuation, but is not a surgical emergency [963,964] unless there is colonic atonia and the patient is in danger of ruptured bowel, or a similar life threatening condition—in which case the condition is the emergency, not the cauda equina.

Compressive abscesses caused by CDs are a surgical emergency. However, in non-compressive CES caused by acute CD leptomeningitis, therapy consists appropriate anti-infective therapy, ± evidence-based immunomodulators (i.e., etanercept, infliximab, thalidomide) if available or on a clinical trial, ± dexamethasone.

A similar approach would be appropriate for most primary or metastatic cancer to the leptomeninges causing CES. A few malignancies causing compressive CES are extremely cancer-therapy sensitive, thus surgery may not be immediately necessary (e.g., lymphoma, testicular cancer, small cell lung cancer), depending on institutional practices. Most solid tumors causing compression will require surgery. Cancers causing CES or other nerve dysfunction confined to the SAS can be treated with intra-CSF chemotherapy or cranial-spinal radiation therapy.

When caused by LMF, we found no published randomized studies demonstrating open decompressive surgery or steroids are superior to supportive care. Results of recent use of intra-SAS neuroendoscopic lysis of adhesions are awaited. Educational emergency "self-help plans," appropriate supplies (self-catheter for urine, gloves for manual stool removal), training on how to use them, and a steroid pack if they respond to steroids helps patients manage at home [961,963,965].

## Treatment

**Lack of successful treatment for acute, iatrogenic non-CD leptomeningitis.** There were 3233 cases of iatrogenic acute leptomeningitis in 108 reports, occurring hours to days after intervention. Some were large clinical trials or population studies. Yet no trials report successful interventions to stop permanent LMF once acute symptoms have started. Although patient groups believe steroids started within days of initial symptoms prevent permanent LMF, this is not supported by peer-reviewed evidence. Spinal fusion surgeons strongly discourage use of steroids and even non-steroidal anti-inflammatory drugs post fusion due to proven risk of decreased fusion. Fusion outcomes are sometimes even used for quality reporting. Further discourse within spinal intervention specialties is needed.

**Promising biologic or immune therapies in TBM and NCC trials.** A 2019 study shows progress is being made in LMF caused by NCC in immigrants to the USA with apparent long-standing infections. Although 41% required permanent shunt for symptomatic hydrocephalus, improvement of other neurologic deficits was seen when immunosuppressants (methotrexate, steroids, etanercept) were added to extended term cysticidal therapy, with 14/33 (42.4%) having no post-treatment sequelae at 4.2 years. The remaining 57.6% had residual neurologic symptoms with 8/33 (24.2%) disabled, 8/33 (24.2%) episodic symptoms, focal neurologic deficits and seizures, and 5/33 (15.2) with mild intellectual impairment [141]. Although the study number is small, the complete recovery rate in this study is substantially higher than large population studies in the PSR.

Also included are the first reports of the effectiveness of thalidomide use in children with base of brain TBM, presenting with blindness or loss of visual acuity, CN and focal nerve deficits, and other brain and spine complications of TBM. Without surgery, "satisfactory clinical results" with radiologic improvement of DLMs was observed in 4/4 patients in the first report and 37/38 in the expanded cohort [966].

Methotrexate is used as a glucocorticoid-sparing agent to diminish the long-term AE associated with prolonged steroid administration. Etanercept is a tumor-necrosis factor inhibitor which decreases proinflammatory cytokines and is most often used in autoimmune diseases. Thalidomide is an antiangiogenic drug currently used in multiple myeloma and transplant related graft vs host disease, classified as an immunomodulatory drug with a black box warning for fetal teratogenicity.

**Base of brain and CN deficits.** There were 83 sources containing 815 patients with cranial nerve symptoms caused by DLMs, accounting for 6.4% (815/12721) of patients in the PSR. Studies spanned from 1924–2019. Patients without a CD diagnosis or who failed therapy for TBM, or other organisms received surgery. Most base of brain reports (34.4%) involved visual

changes due to optochiasmatic arachnoiditis (OCA). Unexpectedly, 21.4% (60/280) cases were iatrogenic due to (now obsolete) muslin wrapping of aneurysms in the past. Due to the wide range of dates, changes in surgical and medical management, and development of MRI, a subset cohort of the most recent 20 years is planned.

**Open and endoscopic lysis of adhesions.** There were 45 articles containing 1258 patients dated from 1924–2019 treated with open, microsurgical, and endoscopic lysis of adhesions, of which 972 (77.3%) were identified as having LMF. Most studies reported early improvement ± new symptoms. Neurologic reporting was incomplete in all studies and all time points. Multiple surgical approaches were taken. Patients who survived had differing post-operative rehabilitation. Relapse or death from complications of disability was high. Meaningful outcome or quality of life differences was not clear.

Thirteen included contemporary articles on endoscopic lysis of adhesions (6 cranial, 7 spinal) demonstrate increasing ability to navigate the SAS, fenestrate adhesions, and perforate arachnoid cysts since 1991. A recent publication on 10 patients with extensive cervical and thoracic adhesions treated with a multimodality approach concludes the approach is feasible, particularly in early disease prior to spinal cord damage [967]. Follow-up is short. None of the 3 neurosurgery authors would perform open surgical lysis of adhesions based on current data. Future developments in minimally invasive lysis of adhesions are promising.

**Spinal cord stimulation (SCS).** Table 6 shows references for 10 articles on SCS, ranging from 1976–2016. Of the 175 total patients with SCS implanted, 72% (n = 126) of had documented LMF or arachnoiditis. Pre-implantation use of opioids, number of repeat procedures, and complications were not consistently reported. However, when reported, a small number of patients counted for the majority of the re-implantations in the studies. Follow up time ranged from 6 months to 10 years. Studies excluded patients thought to be psychologically inappropriate or demonstrating drug-habituation.

The overall response rate was 64.3% (81/126 DLM patients) of this carefully chosen group, with demonstration of reduction of pain and >7 patients with prior disability able to return to work at some level. In DLMs patients receiving SCS, 15.1% (n = 19) achieved complete resolution of pain, including several patients who had resolution of neurologic changes and ultimately returned to work. Another 49.2% (62 patients, >9 of whom were on pre-implantation opioids) achieved a significant partial response (pain decreased 50–99%). There was no change in 23.8% (30 patients, >2 on opioids), and worsening of symptoms in 10.3% (13 patients, >1 on opioids). There were >52 re-implantations, with 10.3% (13/126) patients accounting for all revisions. Additional complications included 11 device infections, 1 iatrogenic paraplegia, and 5 unrelated deaths.

Some vocal patients within the FB community vehemently oppose SCS, stating further instrumentation of the LM will worsen the disease process. This PSR showed worsening of pain or new neurologic deficits in only 13.3%. The included articles discuss "adequate psychiatric screening" and the role of concurrent opioid use in SCS outcome. "Appropriate" use of concurrent opioids was not correlated to lack of response in this PSR, though patients determined to demonstrate addictive behavior or psychologic "problems" (psychologic difficulties, drug habituation, issues of secondary gain) were excluded.

The patient-reported practice of mandatory weaning off opioids prior to implantation of an SCS in some surgical practices is a contentious topic in patient communities. This PSR was not designed to address this practice, however as with several prescription drugs (hypertensives, psychiatric drugs, steroids), slow tapering is required for endogenous systems to acclimate to withdrawal of exogenous drugs without rebound symptoms, some of which can be life threatening. Intractable neuropathic pain can be unbearable without adequate treatment. There is no evidence to support weaning psychologically fit patients off needed opioids for SCS to be

successful. We recommend evidence-based or evidence-generating approaches to pain management prior to SCS. SCS technology has advanced significantly in the past several years and is an important component of managing intractable pain and some neurologic deficits from DLMs in appropriately chosen patients.

**Rehabilitation.** After critical review and removal of poor studies, the remaining three studies had four patients using neuro-mobilization in LMF or LM ossificans. Two patients with LMF had measurable decreases in pain. Two patients with LM ossification were wheelchair bound, but after surgery and intensive rehabilitation were wheelchair free for at least the 4 years [724,852,853].

Neural flossing, as recommended by Mayo Clinic Spine Rehabilitation (author consultation), is not found in structured databases, but there are multiple online resources. We use and teach evidence-based mind-body techniques. The hypothesis is slowly mobilizing nerves and tissues also break fibrotic strands. This approach is best accomplished in the setting of a multidisciplinary mind-body team. The American College of Physicians has published systematic review evidence indicating yoga, tai chi, meditation, acupuncture, and massage increased activity and decreased chronic pain [968]. We recommend patients try any or all of these, pick what works and exercise regularly with slow progression. Swimming is also great exercise. Patients may flare at first, but they should continue slowly through the flare.

**Neuropathic pain (NPP).** The 2017 review above evaluated nonpharmacologic therapies for low back pain, was not specific to patients with LMF. However, the PPI team uses and endorses them for improving activity and decreasing disease distress.

There are several evidence-based guidelines for pharmacologic treatment of NPP, not included in this PSR. As discussed above, 64.3% (81/126) DLM patients has a response to SCS, with 15.1% CR, many stopping opioids and even returning to work. For patients not controlled on high doses of opioids or looking for symptom improvement, SCS is an important adjunct with newer modalities hopefully enhancing response.

Cordotomy and rhizotomy are generally used as a last resort for intractable pain. There were 7 articles with 33% (69/209) DLM patients. For areas below the cervical nerves, rhizotomy failure rate is high, up to 73% in patients. Cordectomy was used in those with intolerable and intractable LE pain and loss of function wishing to preserve UE function and improved quality of life. Of the 15 performed, 1 patient had continued pain and 3 worsening LE spasm. Most responders demonstrated improvement of LE motor and sensory function.

**IT treatments of DLMs symptoms.** Both IT urokinase and hyaluronidase have been used to treat existing LMF. Urokinase was used in preterm infants with intraventricular hemorrhage to prevent hydrocephalus, showing those receiving low dose urokinase required fewer shunts [841]. Three studies showed IT hyaluronidase benefitted OCA, spinal LMF, and noninfective LMF [189,576,842]. Many early articles attempted IT treatment of TBM and chronic fungal infections with traditional systemic antifungal drugs (amphotericin B}, universally causing worsened LMF.

**Successful CSF exchange after IT methotrexate overdoses and toxic leptomeningitis.**
Two boys received IT methotrexate overdoses of 240 mg instead of 12 mg on the same day due to pharmacy error. Both developed severe acute leptomeningitis with seizures and mental status changes. Glucarpidase was not on site. Untreated, death or diffuse LMF with progressive leukoencephalopathy was predicted [291]. CSF exchange was performed 5 hours after overdose by removing fixed amounts of CSF via LP catheter, then replacing it with warmed normal saline. Treatment was successful in one boy but was incomplete in the other. Glucarpidase was administered IT 11 hours later. Only short-term memory loss in the child with incomplete exchange is reported at three years follow-up. Authors state if patients have an implanted

ventricular Ommaya, the procedure can be performed as a ventriculolumbar perfusion for CSF drainage.

The report reviewed 8 previous publications with 13 patients receiving successful CSF exchanges, though none were identified by our searches, highlighting neurologic terminology inconsistencies which has obscured knowledge of this apparent life-saving procedure in toxic IT injections. It also highlights limitations of using only structured database searches to reach conclusions.

We include this report as there have been accidental IT overdoses of other drugs for which there is no antidote resulting in death or permanent severe disability. Knowing CSF exchange can be safely performed may provide a glimmer of hope to an otherwise dire situation.

**Other interesting therapies for possible clinical trials.** The PSR returned one report on "cerebrolysin" (not available in the USA) widely used in a few countries [850]. According to the article "cerebrolysin is a peptide solution containing free amino acids biologically active peptides" including fibroblast growth factor, brain-derived neurotrophic factor, glial-derived neurotrophic factor, TGF beta, and insulin-like growth factor 1. Of the 3/6 patients who could have had DLMs (2 viral meningoencephalitis; 1 traumatic avulsion RUE), all 3 had "remarkable recovery with acceptable residuals." Further study is needed.

Metformin animal studies and reviews were not eligible for inclusion in the PSR. However, one member of the PPI was placed on metformin during data extraction for the PSR. Within 72 hours, they experienced a severe flare of the exact NPP, sensory and motor loss experienced in areas that had regained function and been quiescent for years. We immediately performed a literature search. Instead of finding metformin caused neuroinflammation, the search returned 4 animal articles indicating metformin reduces brain damage due to ischemia, decreases NPP, has anti-neuroinflammation properties, and promotes NSC activation [969–972]. The very anxious PPI member agreed to a one month trial. Within 3–4 weeks the flare stopped. Their neurologist documented improvement in sensory dysesthesia and reflexes from the previous exam prior to the start of metformin, which has continued. News spread quickly to FB groups. Now metformin is currently widely used, ending our plan for a placebo-controlled trial. Recent large metformin studies have demonstrated metformin is inexpensive with a good safety profile in the majority of people, even without hyperglycemia [973].

Not in the PSR, a 2022 critical review of metformin discusses the potential benefits, postulating mechanisms in a variety of positive clinical trials. *In vitro* studies shows restoration of autophagy, and activation of NSC among other cellular events. Induced B12 deficiency due to malabsorption occurs in 6–30% of chronic users, indicating the need for monitoring and sublingual replacement, but metformin use is not associated with hypoglycemia in the euglycemic population [973]. Placebo-controlled clinical trials are needed to determine efficacy in DLMs.

## Treatments without sufficient evidence

We recommend against use of the 2018 Tennant cocktails of "neuro-steroids and neurohormones" due to lack of studies in humans and questioned safety of supplements [893]. Since those protocols were written, a number of advances in targeted drugs and the molecular pathways involved in LMF have been made with better safety profiles for clinical trials.

We also recommend against the use of neural stem cells (NSC) off a clinical trial. The PPI team personally is aware of hundreds of LMF patients who have had these therapies at least once, many repeatedly. Currently, only the financially fortunate have access. A well conducted clinical, trial if positive, could lead to insurance coverage and decreased disparities. Positive trials of NSC in DLMs have not been published and harm is documented [974]. Evidence exists of stem cell culture supernate having a significant, though temporary impact on neurology

recovery in spinal cord injury patients [975]. We found no evidence NSC can induce recovery LM cells, which come from a different embryonic lineage.

Hundreds, if not thousands of DLMs patients have tried low dose naltrexone(LDN) based on Tennent protocols and multiple sclerosis websites. There is sufficient data of a high safety profile for LDN in patients not taking opioids. The theory is low doses bind opioid receptors on immune modulating cells causing activation as well as increasing response to opioids by upregulating receptors [976]. No data appeared in our PSR. There is insufficient evidence for any conclusion.

Curcumin, a known anti-neuro-inflammatory agent that impacts glial cells in did not show up in the PSR. A report in rat SAH models demonstrates high doses of curcumin ameliorated the post inflammatory response and subsequent LMF after induced SAH. This striking response deserves further investigation as SAH has devastating LMF consequences. At therapeutic doses in humans it does not provide the neuroprotective, anti-neuroinflammation, or antioxidant affects seen in preclinical studies [977]. Doses nearing the toxic range might be necessary due to poor absorption [978]. Serious liver toxicity has been noted. Newer formulations are entering clinical trials.

## Brief molecular biology primer

Though not included in results syntheses, we prospectively included pre-clinical research for future directions and were surprised at the advances in LM composition and functions. Of >1000 reviews and bench research studies going back to 1832, we collected and reviewed pertinent original non-human studies published after 2010. With the discovery of a glymphatic system in the dura and LM in 2015 there has since been exponential growth in understanding CSF flow and its relationship to both CNS and peripheral organs [906].

Current understanding is the SAS CSF plays a critical role in the process of providing oxygen and nutrients to neurons and removal of metabolic waste, in addition to housing multiple macrophages and NSC. Publications document the critical role the glymphatics play in maintaining normal brain function through aging as well as in neuroinflammation, injury, and cancers. The evolving framework explains the clinical syndromes which involve inflammation or disruption of LM architecture, CSF flow, and neurovascular units. A recent molecular and cellular biology review confirms this from scientific and animal study points, discussing clinical scenarios [900].

Several articles discuss the importance of the LM stem cell niches in migrating to and repairing CNS damage. We were unable to find evidence NSC can regenerate the LM or SAS, but studies have shown use of NSC or NSC supernate improve symptoms, function, and pathologic appearance of spinal cord lesions given through different routes in animal models. Although SAH generally leads to LMF and serious disability, preliminary animal studies with NSC may produce options post SAH, but data are early [979].

Because hrBMP-2 is used extensively in spine surgeries, we attempted to elucidate the mechanism by which rhBMP-2 might cause DLMs. BMP-2 was first identified because of its role in bone formation but there are multiple BMPs which participate in a wide variety of metabolic pathways for multiple cell types. With the exception of BMP-1, all other BMPs are part of the transforming growth factor (TGF-β) superfamily of secreted growth factors essential for cellular communication [980,981]. The TGF-β family contains >30 growth factors, all involved in just two SMAD signaling pathways (SMAD2/3 or SMAD 1/5/8) [982].

The apparent cause of rhBMP-2 DLMs is initiation of the SAS inflammatory and fibrotic cascades. A 2016 review of the clinical and pre-clinical side effects of rhBMP-2 discusses the intense inflammatory responses in addition to radiculitis, ectopic bone, osteoclast activation

and other complications by activation (IL(interleukin)-1β, IL-6, IL-10, IL-17, IL-18, tumor necrosis factor-α, potentiation of nuclear factor-kβ ligand, induction of RANKL, and dose-dependent induction of PPARγ expression) [983]. Animal studies with rhBMP-2 have demonstrated activation of members of TGF-β1 decrease the ability of NSC to repair CNS damage as well as increase LMF [908,984–986].

The complete BMP/ TGFβ pathways are beyond the scope of this paper. Indeed different BMP/ TGFβ ligands can assemble identical Type I or II receptors that do not result in the same signaling pathway [980]. While hrBMP-2 is implicated in causing fibrosis, BMP-7 appears to decrease TGF-β-associated tissue fibrosis and enhance recovery of cardiac, pulmonary, and renal fibrosis [984]. We are scratching the surface of these biologically potent molecules. We strongly encourage spine surgeons and interventionalists to become active in initiating and supporting clinical trials in what will become a new field of study—prevention of permanent damage to the SAS and LMs, perhaps starting with early use of immune modulators showing activity in TBM.

## ICD 11 proposal for consistent naming convention

As we progress in targeted therapy, improving outcomes depends on utilizing accurate, consistent terminology and unique International Classification of Diseases (ICD) coding to establish databases for appropriate treatments, outcomes analysis, and eligibility for clinical trials. Going forward, it will be critical to classify and capture different phases and forms of DLMs. In the real world DLMs represents several different pathologies for which there currently is no ICD10 category or "parent" for "Diseases of the Leptomeninges." The prognosis is not the same for all DLMs. Causes, symptoms, current and proposed future treatments are likely to be different.

We proposed to the WHO complex hierarchical system changes for ICD11, as shown in Fig 5 for creation of a new chapter to be named "Diseases of the meninges." We will resubmit the proposal upon demonstration of increased medical awareness the need for these changes. Hopefully clinical trials will produce therapies that when used in the acute phases will prevent permanent LMF progression.

## Certainty of body of evidence

We are confident the study results represent overall certainty of evidence in our primary conclusions and recommendations and will not be changed by others repeating this SR using the same time frame and our published methodology. We are hopeful future SR with later dates will have different results because of prevention, global disease treatment equity, and medical innovation

## Implications for practice and future research

Based upon the scientific rigor of this review, we make several suggestions. First is dissemination of the information in this report to the medical and patient communities, including enhanced patient education efforts. Table 9 contains some possibilities for future research, but there are likely unpublished therapeutic candidates to consider. We propose using ICD11 to clarify and unify terminology and disease differences. Prevention and treatment for acute disease should be a priority. Finally, we must do a better job training radiologist to recognize the presence of DLMs on MRI's, and we must do better bringing the outcomes in N-OECD up to OECD standards, particularly in treatment of curable diseases.

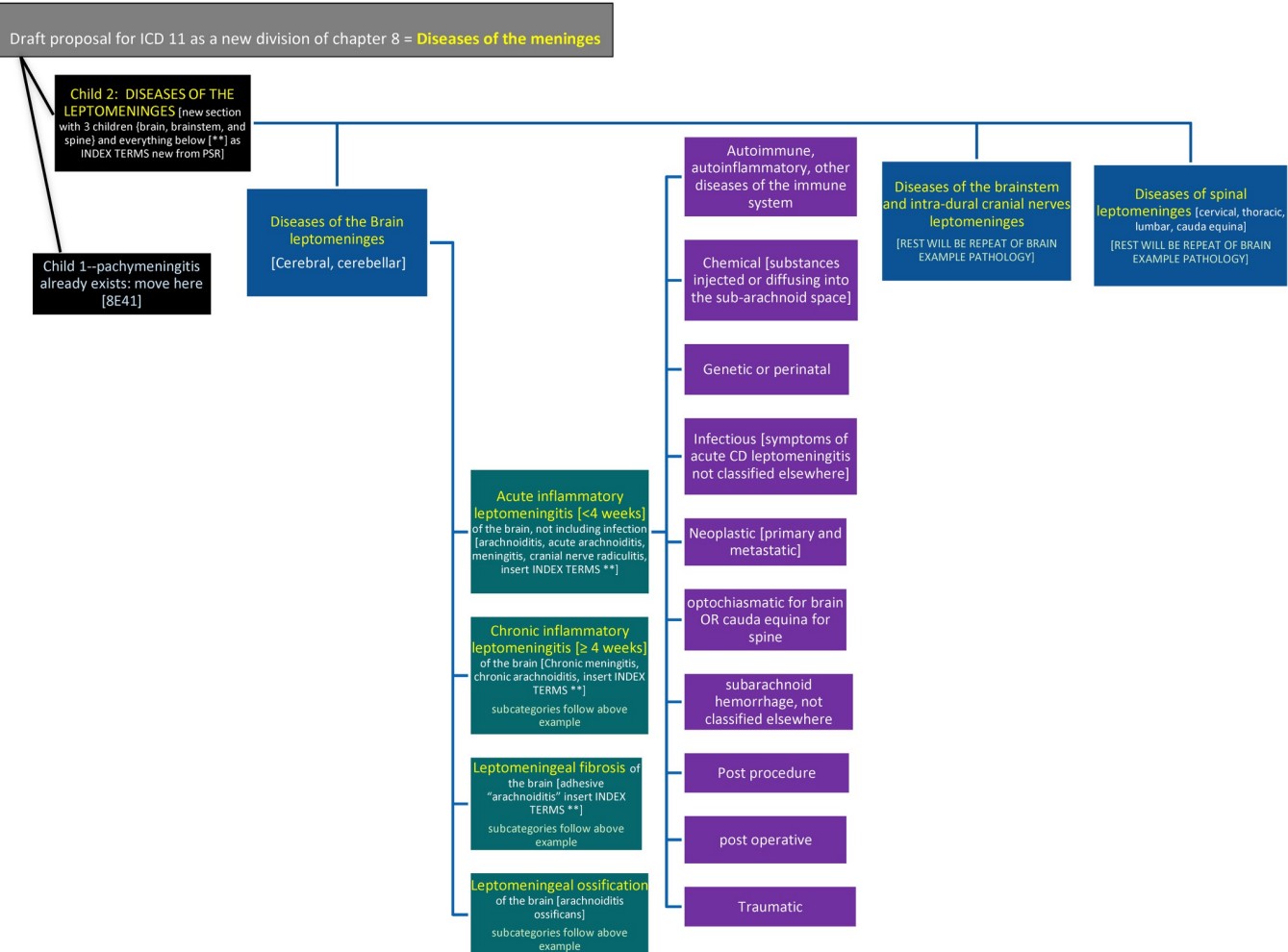

**Fig 5. Proposed ICD11 Classification system for a new section named "Diseases of the meninges".** The current version of ICD 11 does not contain a section for meningeal diseases, although there are scattered places where this complication appears as a complication of a specific disease. As many disease entities cause the same pathologic changes and symptoms, progress in quantifying LMDs, clinical course and outcomes, and eligibility for clinical trials will only be possible with consistent coding. We have proposed the hierarchical system below, with "grandchildren" being brain, brainstem, and spinal involvement of the LM. The example "generation 4 and 5" in green and purple would be repeated for each of the 3 main grandchildren. This proposal will be resubmitted to WHO, with all input welcome.

**Table 9. Possible treatment ideas for clinical trials from the PSR and subclinical studies.**

| Studies in humans with DLMs reported or underway | Studies in humans with other diseases reported or underway | Animal study results, no human data | Molecular or cellular biology studies suggestive of effect |
|---|---|---|---|
| Thalidomide | NSC | metformin | metformin |
| Etanercept + steroids | cerebrolysin | curcumin | |
| Methotrexate + steroids | Low dose naltrexone | | |
| Anakinra + steroids | NSC supernate | | |
| Endoscopic surgery for lysis of adhesions | | | |
| Infection-specific therapy | | | |
| Bladder stimulation | | | |
| Spinal cord stimulation | | | |

## Conclusion

We conclude:

1. Clinically, the LM constitute an organ with functions critical to the development and maintenance of the underlying CNS, as well as transiting neurons that comprise the peripheral nervous system, confirming preclinical research findings.

2. The SAS porosity and unique cellular components (NSC, macrophages, etc.) must be preserved for normal CSF flow and CNS function, and is the anatomic area involved in "arachnoiditis, chronic meningitis," as well as the other leptomeningeal pathologies ± concomitant CNS pathology.

3. A spectrum from acute injury to complete fibrotic destruction occurs, the course of which differs by defined causes and can be captured by proposed ICD 11 terminology since coding accuracy identifies patients for providing treatments and measuring best outcomes and quality of life in searchable databases.

## Supporting information

**S1 Checklist.**
(DOCX)

## Acknowledgments

We thank: the Mayo Clinic Library and librarians for donation of time and resources; the modified d-Delphi team, named with written permission (Subhash Patel MD, Tiina Säynäjoki MS (Chief Data Integrity Officer), Marlisa Griffith RN BSN, Tracy Kruzick MD, Sarah Fox MBBS, Robert West PhD, Allison Tucker RT, Jill Ackerman MD, Rich Allen MD); the numerous other patients contributing time searching resources in the early phases of the PSR; the experts contributing content to the paper, named with written permission [Christopher Cohen PhD for Fig 4 pictures; Francesco Fornai MD for Fig 4 pictorial; Dieter Hoelzer MD PhD, President European Hematology Association and Director Emeritus, Department of Medicine, University of Frankfurt, Germany (personal communication); Richard Larson MD. Director Hematologic Malignancy Clinical Research Program, The University of Chicago (personal communication)]; neurosurgeon Charles V. Burton MD for multiple contributions prior to his death in December 2020.

## Author Contributions

**Conceptualization:** Carol S. Palackdkharry, Erin Dienes, Mohamad Bydon, Michael P. Steinmetz, Vincent C. Traynelis.

**Data curation:** Carol S. Palackdkharry, Stephanie Wottrich, Erin Dienes, Michael P. Steinmetz, Vincent C. Traynelis.

**Formal analysis:** Carol S. Palackdkharry, Stephanie Wottrich, Erin Dienes, Mohamad Bydon, Michael P. Steinmetz, Vincent C. Traynelis.

**Funding acquisition:** Carol S. Palackdkharry, Vincent C. Traynelis.

**Investigation:** Carol S. Palackdkharry, Stephanie Wottrich, Erin Dienes, Mohamad Bydon, Michael P. Steinmetz, Vincent C. Traynelis.

**Methodology:** Carol S. Palackdkharry, Stephanie Wottrich, Erin Dienes, Mohamad Bydon, Michael P. Steinmetz, Vincent C. Traynelis.

**Project administration:** Carol S. Palackdkharry, Erin Dienes.

**Resources:** Carol S. Palackdkharry, Stephanie Wottrich, Erin Dienes, Vincent C. Traynelis.

**Software:** Carol S. Palackdkharry, Stephanie Wottrich, Erin Dienes, Vincent C. Traynelis.

**Supervision:** Carol S. Palackdkharry, Erin Dienes, Mohamad Bydon, Michael P. Steinmetz, Vincent C. Traynelis.

**Validation:** Carol S. Palackdkharry, Stephanie Wottrich, Erin Dienes, Mohamad Bydon, Michael P. Steinmetz, Vincent C. Traynelis.

**Visualization:** Carol S. Palackdkharry, Stephanie Wottrich, Erin Dienes, Michael P. Steinmetz, Vincent C. Traynelis.

**Writing – original draft:** Carol S. Palackdkharry, Erin Dienes, Mohamad Bydon, Michael P. Steinmetz, Vincent C. Traynelis.

**Writing – review & editing:** Carol S. Palackdkharry, Stephanie Wottrich, Erin Dienes, Mohamad Bydon, Michael P. Steinmetz, Vincent C. Traynelis.

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
