## [Decision Letter · Decision Letter 0]

17 Jun 2022

PONE-D-22-02599The leptomeninges as a critical organ for normal CNS development and function: first PPI systematic review of arachnoiditis (chronic meningitis)PLOS ONE

Dear Dr. Palackdkharry,

Thank you for submitting your manuscript to PLOS ONE. After careful consideration, we feel that it has merit but does not fully meet PLOS ONE’s publication criteria as it currently stands. Therefore, we invite you to submit a revised version of the manuscript that addresses the points raised during the review process.

We appreciate your patience. The reviewers' comments are provided below. Please ensure to address them point by point and format your manuscript as previously discussed. 

We look forward to receiving your revised manuscript.

Kind regards,

Panagiotis Kerezoudis, M.D., M.S.

Academic Editor

PLOS ONE

Journal Requirements:

Dr. Palackdharry: none 

Stephanie Wottrich: none.

Dr. Dienes: none.

Dr. Bydon: none

Dr. Steinmetz: Elsevier: royalties; Globus: consultant; Medtronic: honorarium; Zimmer Biomet: royalties.

Dr. Traynelis: Medtronic: consultant, royalties; NuVasive: consultant 

3. We noted in your submission details that a portion of your manuscript may have been presented or published elsewhere. Please clarify whether this conference proceeding was peer-reviewed and formally published. If this work was previously peer-reviewed and published, in the cover letter please provide the reason that this work does not constitute dual publication and should be included in the current manuscript."

5. We note that you have referenced (ie. Bewick et al. [5]) which has currently not yet been accepted for publication. Please remove this from your References and amend this to state in the body of your manuscript: (ie “Bewick et al. [Unpublished]”) as detailed online in our guide for authors

6. Please include a caption for figure 5.

Reviewers' comments:

Reviewer's Responses to Questions

**Comments to the Author**

1. Is the manuscript technically sound, and do the data support the conclusions?

Reviewer #1: Yes

Reviewer #2: Yes

Reviewer #3: No

Reviewer #4: Yes

2. Has the statistical analysis been performed appropriately and rigorously? 

Reviewer #1: Yes

Reviewer #2: N/A

Reviewer #3: N/A

Reviewer #4: Yes

3. Have the authors made all data underlying the findings in their manuscript fully available?

Reviewer #1: Yes

Reviewer #2: Yes

Reviewer #3: Yes

Reviewer #4: Yes

4. Is the manuscript presented in an intelligible fashion and written in standard English?

Reviewer #1: Yes

Reviewer #2: Yes

Reviewer #3: Yes

Reviewer #4: Yes

5. Review Comments to the Author

Reviewer #1: Authors conducted a patient and public-involved comprehensive systematic review following PRISMA protocol. The manuscript is well written.

Resolution of all figures must be improved in revision submission.

Reviewer #2: The review by Palackdkharry et al. seems to be an interesting approach to highlight arachnoiditis and related several neuronal disorders that fall under this category. Their detailed review is clinically relevant in the field of neurosciences and diseases. However, after reading the review paper thoroughly, I feel that I am not an expert in this field. The paper is better suited to be reviewed from a scientist who is an expert in the clinical research and who has a background in the field of meningitis or related neuronal diseases. I am sorry that I am unable to contribute in reviewing this paper. Although, I liked reading and the authors have explained leptomeninges very well and the whole paper was quite informative. I have only 1 comment and that is: the figures attached in the manuscript were not clear, especially the ones with small fonts. It may be due to the lower resolution size of the figures which got even worse while generating the pdf. Please attach high resolution images in future.

Reviewer #3: Review for article entitled "The leptomeninges as a critical organ for normal CNS development and function: first PPI systematic review of arachnoiditis (chronic meningitis)"

The authors describe a qualitative synthesis of 887 sources used to evaluate pathology that involves the subarachnoid space and the pía mater. They attempt to nullify traditional clinical thinking about diseases of the leptomeninges, answer study questions, and create a unified path forward.

Conclusions include statements about diseases of the leptomeninges being common, the leptomeninges clinical function is critical involving the SAS-pia structure and enclosed cells. Cases of disease of the leptomeninges involve all specialties. Causes are numerous and symptoms predictable and outcomes depending on time to treatment and extent of residual SAS damage. The manuscript is well written.

Major critiques:

The title of the article does not support the conclusions.

Although the authors have done an exhaustive search of the literature, given the nature of diseases of the leptomeninges, the approach of the resulting manuscript is too broad. As such the resulting synthesis is only qualitative. No statistical analysis is possible with the data presented and the conclusions are generalized and subjectively based on the literature review.

The manuscript would benefit from a more narrowed and specific approach to arachnoiditis or specific aspects of diseases of the leptomeninges.

Additional concerns:

Legal studies (n=7) were included. These will have an inherent bias of the author of such work

I could not locate the PSR protocol on the Arcsology.org website

Greater than 100 years span for sources creates greater difficulty in analyzing results given the advancement in medical and surgical techniques over this time frame.

Reviewer #4: The authors report an excellent and comprehensive systematic review on arachnoiditis including data from 887 studies and over 12,000 patients. They highlight that communicable diseases are the cause of more cases of “diseases of the leptomeninges” (DLMs) than iatrogenic causes. They also report an association between use of rhBMP-2 and DLMs. They found that only 1.1% and 0.2% of permanent DLMs are associated with spinal injections and neuraxial anesthesia, respectively. They comprehensively cover DLM-related causes, diagnostics, symptoms, treatments, and propose a new classification scheme to study DLMs in the future. This warrants eventual publication with the following comments/concerns:

    ⁃    PPI should be spelled out in the title

    ⁃    There are several distracting typos throughout (e.g., lines 7-8 in Objectives, Line 40, Line 62, Line 170, lines 209-212, Line 262, Lines 267, Lines 270-272, Line 423, Line 497, Line 677, Lines 781, Line 803). It would greatly improve readability of this excellent manuscript of these were fixed.

    ⁃    Lines 111, “we added 632 patients with neurological AE consistent with DLMs.” Please provide more detail. How were these adjudicated?

    ⁃    In Line 223, is the number of DLM patients 12724 or 12624 as in the abstract?

    ⁃    Table 7 totals are cut off and not viewable.

    ⁃    Table 8 is also cut off for my review

    ⁃    Please clarify, in lines 612-622, the specific chemotherapeutic and immunomodulators used in the treatment of compressive abscess.

    ⁃    The paper reports the dissemination of metformin throughout the PPI FB group. Were there any adverse effects related to metformin in the FB group?

    ⁃    In the discussion of the observed association between rhBMP-2 and DLMs, what are the potential mechanisms for this link? Do these potential mechanisms have any implications for changes in clinical care? Is the increased risk mediated by other factors in surgeries utilizing BMP (e.g., which are often revision surgeries)? BMP is an important surgical adjunct so the others should address these additional items.

    ⁃    Spinal cord stimulation is a very important treatment option for patients with DLMs, the authors may consider devoting more extensive specialized discussion on this topic.

6. PLOS authors have the option to publish the peer review history of their article (what does this mean?). If published, this will include your full peer review and any attached files.

Reviewer #1: **Yes: **Masum Rahman

Reviewer #2: No

Reviewer #3: No

Reviewer #4: No

---

## [Author Response · Author response to Decision Letter 0]

19 Jul 2022

Below is a list of revisions and/or rebuttals without revisions:

1. Throughout the abstract, paper, and tables, the total number of study and DLMs patients has increased 61 patients to 133261 and 12721, respectively. This is due to an extraction error in the spinal cord stimulation patients discovered in writing the new section [North published 2 papers in the same year and the case report was accidentally extracted instead of the study with 61 patients]. The resulting numbers further support conclusions, but do not change overall study percentages due to the large number of patients in the study. 

2. The second sentence of the conclusion of the abstract was rearranged since the word “function” was repeated.

3. From Dr. Panagiotis Kerezoudis, Academic Editor

a. The abbreviation “PPI” was spelled out in the title

b. Author bylines and affiliations were corrected according to the PLOS formatting guidelines

c. Author symbol legend was corrected according to the PLOS formatting guidelines

d. All figures were remade in high resolution 600 dpi .tiff, uploaded to the PACE Tool, which converted down to 300 dpi automatically and with the .tif extension. These did not give good resolution at smaller sizes, so we are uploading the new high resolution 600 dpi.tiff files

e. Tables 5-8 were reformatted to fit the standard layout

f. Tables and captions are no longer embedded in the text. All tables with legends were moved to the end of the text, prior to references, and the reference numbers appropriately changed.

g. Figure legends were moved to after the references.

h. Figure 2 has been referenced [as all pictures were published in a journal article]

i. Competing interests have been shared. This does not alter our adherence to PLOS ONE policies on sharing data and materials. 

j. Competing interests have been updated and there are no changes

k. The data was presented as a podium poster presentation AANS in July 2021. The poster was peer-reviewed but not published as part of the conference proceedings. Therefore, no work constitutes dual publication, and all data should be included in the manuscript. 

l. All data used for the manuscript has been made available in Table 5. We will change the data availability statement to reflect that all relevant data is available within the manuscript. 

m. We are unsure of the source of the comment regarding inclusion of Bewick et al [5]. We do not have that citation in our Zotero database or cited anywhere in the article. There is no reference to this citation in our document.

n. A caption for Figure 5 has been provided after the references. 

4. Response to reviewers # 1 & 2: All figures were changed to high resolution [300 dpi] using the PACE tool after retaking image shots. We thank the reviewers for their time and comments.

5. Response to reviewer #3:

a. We believe the title of the article supports the conclusion that the LM are a distinct organ. There is a large body of animal and bench research supporting this concept. Our exhaustive PSR in humans not only supports our conclusion but opens a new field in neuroscience to prevent infectious, autoimmune, toxic, and iatrogenic damage that leads to fibrosis of the critical structures that traverse the SAS and support remodeling of the CNS throughout life. 

b. We agree a quantitative analysis of the broad range of data is not possible, which is why we did not attempt one. Statistical analysis is possible with certain portions of the data, which have already been performed and reported. We have added additional studies not included in those analyses.

c. We do not believe our conclusions are subjective, but solidly based on our data as well as a wealth of pathologic and pre-clinical data. 

d. The broken links to the published PSR have been fixed. 

e. We have provided substantial evidence that author-industry relationships bias published literature and legal data increase knowledge of possible harms. When devices or techniques cause harm, settlements often prevent publication of those harms and thus the literature can overestimate benefit. These concepts are discussed and referenced in the article.

f. We agree the field of medicine has changed over the nearly 200-year span of this PSR, thus we did not attempt make conclusions impacted by changes in imaging and surgical techniques. Going forward, with the information we have integrated, it will hopefully be easier to create and share innovations. 

6. Response to reviewer #4: we thank this reviewer for their careful review and helpful comments.

a. PPI was spelled out in the title and all noted typos were fixed.

b. We added details of how the AE consistent with DLMs were adjudicated to come up with the resulting number of 632 added events in the “iatrogenic DLM” section

c. Table width was fixed for tables 5-8

d. Denominators were changed to be consistent throughout the manuscript. The new denominator is 12721. There were 50 patients with SCS implanted for “lumbar meningeal fibrosis” which were added when we re-extracted all 10 SCS articles. 

e. We added in the immunomodulators, and anti-infective agents used to treat compressive abscesses. 

f. We answered the question of AE observed in non-DM metformin users. FYI, it was well tolerated, as evidenced by a 2022 review now cited in the discussion [no data were included as it was out of the cutoff range but used for discussion].

g. We added in the proposed mechanisms for rhBMP-2 DLMs in the molecular biology section. There is substantial information on many other AE associated with this BMP2 [non- DLMs] which are in the references. We added a sentence about reduced doses in the iatrogenic section, as well as discuss the role of multiple surgeries. 

h. We are immensely grateful reviewer 4 pointed out we left out the section on SCS, despite specifically planning for it. We agree it is a critical issue and added in a section under “treatment,” as well as rearranged the treatments from practical at this moment to theoretical and unproven.

---

## [Decision Letter · Decision Letter 1]

1 Sep 2022

The leptomeninges as a critical organ for normal CNS development and function: first patient and public involved systematic review of arachnoiditis (chronic meningitis)

PONE-D-22-02599R1

Dear Dr. Palackdkharry,

We’re pleased to inform you that your manuscript has been judged scientifically suitable for publication and will be formally accepted for publication once it meets all outstanding technical requirements.

Kind regards,

Panagiotis Kerezoudis, M.D., M.S.

Academic Editor

PLOS ONE

Additional Editor Comments (optional):

Reviewers' comments:

Reviewer's Responses to Questions

**Comments to the Author**

1. If the authors have adequately addressed your comments raised in a previous round of review and you feel that this manuscript is now acceptable for publication, you may indicate that here to bypass the “Comments to the Author” section, enter your conflict of interest statement in the “Confidential to Editor” section, and submit your "Accept" recommendation.

Reviewer #1: All comments have been addressed

2. Is the manuscript technically sound, and do the data support the conclusions?

Reviewer #1: Yes

3. Has the statistical analysis been performed appropriately and rigorously? 

Reviewer #1: Yes

4. Have the authors made all data underlying the findings in their manuscript fully available?

Reviewer #1: Yes

5. Is the manuscript presented in an intelligible fashion and written in standard English?

Reviewer #1: Yes

6. Review Comments to the Author

Reviewer #1: You agree that the authors have satisfactorily addressed your concerns from an earlier round of review and that this manuscript is now suitable for publication.

7. PLOS authors have the option to publish the peer review history of their article (what does this mean?). If published, this will include your full peer review and any attached files.

Reviewer #1: **Yes: **Masum Rahman

---

## [Editor Report · Acceptance letter]

8 Sep 2022

PONE-D-22-02599R1 

The leptomeninges as a critical organ for normal CNS development and function: first patient and public involved systematic review of arachnoiditis (chronic meningitis) 

Dear Dr. Palackdkharry:

I'm pleased to inform you that your manuscript has been deemed suitable for publication in PLOS ONE. Congratulations! Your manuscript is now with our production department. 

Kind regards, 

on behalf of

Dr. Panagiotis Kerezoudis 

Academic Editor

PLOS ONE